# SCALING AND EVALUATING SPARSE AUTOENCODERS

**Leo Gao**[*]   **Tom Dupré la Tour**[†]   **Henk Tillman**[†]   **Gabriel Goh**   **Rajan Troll**
**Alec Radford**   **Ilya Sutskever**   **Jan Leike**   **Jeffrey Wu**[†]
OpenAI
lg@openai.com

## ABSTRACT

Sparse autoencoders provide a promising unsupervised approach for extracting interpretable features from a language model by reconstructing activations from a sparse bottleneck layer. Since language models learn many concepts, autoencoders need to be very large to recover all relevant features. However, studying the properties of autoencoder scaling is difficult due to the need to balance reconstruction and sparsity objectives and the presence of dead latents. We propose using k-sparse autoencoders (Makhzani & Frey, 2013) to directly control sparsity, simplifying tuning and improving the reconstruction-sparsity frontier. Additionally, we find modifications that result in few dead latents, even at the largest scales we tried. Using these techniques, we find clean scaling laws with respect to autoencoder size and sparsity. We also introduce several new metrics for evaluating feature quality based on the recovery of hypothesized features, the explainability of activation patterns, and the sparsity of downstream effects. These metrics all generally improve with autoencoder size. To demonstrate the scalability of our approach, we train a 16 million latent autoencoder on GPT-4 activations for 40 billion tokens. We release code and autoencoders for open-source models, as well as a visualizer.

## 1 INTRODUCTION

Sparse autoencoders (SAEs) have shown great promise for finding features (Cunningham et al., 2023; Bricken et al., 2023; Templeton et al., 2024; Goh, 2016) and circuits (Marks et al., 2024) in language models. Unfortunately, they are difficult to train due to their extreme sparsity, so prior work has primarily focused on training relatively small sparse autoencoders on small language models.

We develop a state-of-the-art methodology to reliably train extremely wide and sparse autoencoders with very few dead latents on the activations of any language model. We systematically study the scaling laws with respect to sparsity, autoencoder size, and language model size. To demonstrate that our methodology can scale reliably, we train a 16 million latent autoencoder on GPT-4 (OpenAI, 2023) residual stream activations.

Because improving reconstruction and sparsity is not the ultimate objective of sparse autoencoders, we also explore better methods for quantifying autoencoder quality. We study quantities corresponding to: whether certain hypothesized features were recovered, whether downstream effects are sparse, and whether features can be explained with both high precision and recall.

Our contributions:

1. In section 2, we describe a state-of-the-art recipe for training sparse autoencoders.
2. In section 3, we demonstrate clean scaling laws and scale to large numbers of latents.
3. In section 4, we introduce metrics of latent quality and find larger sparse autoencoders are generally better according to these metrics.

We also release code, a full suite of GPT-2 small autoencoders, and a feature visualizer for GPT-2 small autoencoders and the 16 million latent GPT-4 autoencoder.

---

[*]Primary Contributor. Correspondence to lg@openai.com.

[†]Core Research Contributor. This project was conducted by the Superalignment Interpretability team. Author contributions statement in Appendix I.

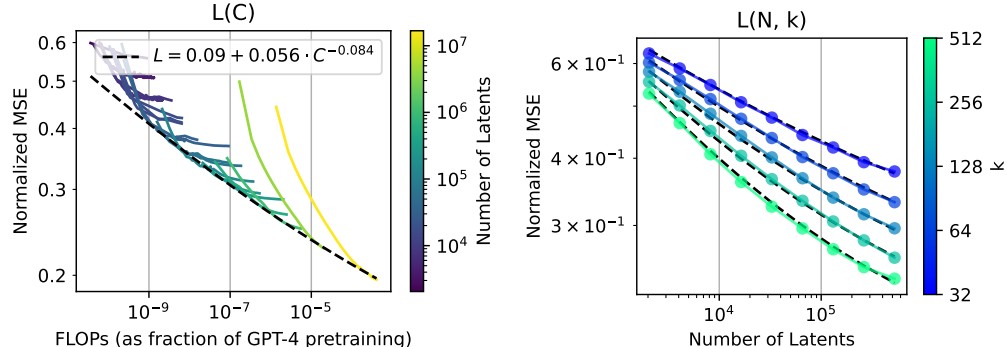

Figure 1: Scaling laws for TopK autoencoders trained on GPT-4 activations. (Left) Optimal loss for a fixed compute budget. (Right) Joint scaling law of loss at convergence with fixed number of total latents $n$ and fixed sparsity (number of active latents) $k$. Details in section 3.

## 2 METHODS

### 2.1 SETUP

**Inputs:** We train autoencoders on the residual streams of both GPT-2 small (Radford et al., 2019) and models from a series of models of increasing size, sharing GPT-4 architecture and training setup, including GPT-4 itself (OpenAI, 2023)[1]. We choose a layer near the end of the network, which should contain many features without being specialized for next-token predictions (see subsection F.1 for more discussion). Specifically, we use a layer $\frac{5}{6}$ of the way into the network for GPT-4 series models, and we use layer 8 ($\frac{3}{4}$ of the way) for GPT-2 small. We use a context length of 64 tokens for all experiments. We subtract the mean over the $d_{\text{model}}$ dimension and normalize to all inputs to unit norm, prior to passing to the autoencoder (or computing reconstruction errors).

**Evaluation:** After training, we evaluate autoencoders on sparsity $L_0$, and reconstruction mean-squared error (MSE). We report a normalized version of all MSE numbers, where we divide by a baseline reconstruction error of always predicting the mean activations.

**Hyperparameters:** To simplify analysis, we do not consider learning rate warmup or decay unless otherwise noted. We sweep learning rates at small scales and extrapolate the trend of optimal learning rates for large scale. See Appendix A for other optimization details.

### 2.2 BASELINE: RELU AUTOENCODERS

For an input vector $x \in \mathbb{R}^d$ from the residual stream, and $n$ latent dimensions, we use baseline ReLU autoencoders from (Bricken et al., 2023). The encoder and decoder are defined by:

$$z = \text{ReLU}(W_{\text{enc}}(x - b_{\text{pre}}) + b_{\text{enc}})$$
$$\hat{x} = W_{\text{dec}}z + b_{\text{pre}} \tag{1}$$

with $W_{\text{enc}} \in \mathbb{R}^{n \times d}$, $b_{\text{enc}} \in \mathbb{R}^n$, $W_{\text{dec}} \in \mathbb{R}^{d \times n}$, and $b_{\text{pre}} \in \mathbb{R}^d$. The training loss is defined by $\mathcal{L} = ||x - \hat{x}||_2^2 + \lambda ||z||_1$, where $||x - \hat{x}||_2^2$ is the reconstruction MSE, $||z||_1$ is an L1 penalty promoting sparsity in latent activations $z$, and $\lambda$ is a hyperparameter that needs to be tuned.

### 2.3 TOPK ACTIVATION FUNCTION

We use a $k$-sparse autoencoder (Makhzani & Frey, 2013), which directly controls the number of active latents by using an activation function (TopK) that only keeps the $k$ largest latents, zeroing the rest. The encoder is thus defined as:

$$z = \text{TopK}(W_{\text{enc}}(x - b_{\text{pre}})) \tag{2}$$

---

[1]All presented results either have qualitatively similar results on both GPT-2 small and GPT-4 models, or were only ran on one model class.

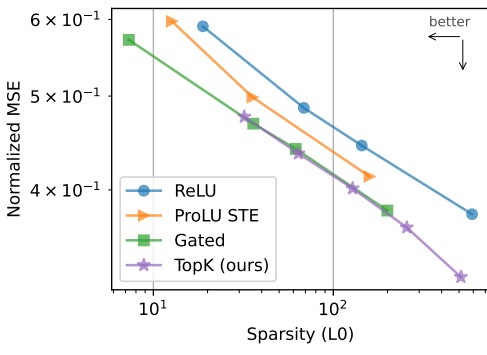 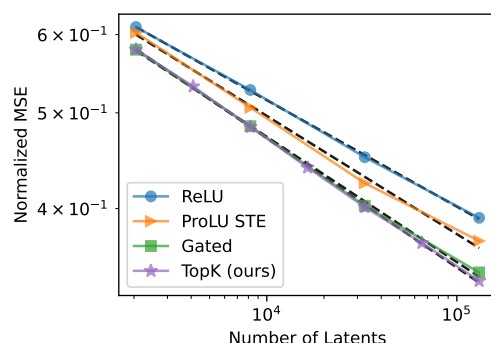

(a) At a fixed number of latents ($n = 32768$), TopK has a better reconstruction-sparsity trade off than ReLU and ProLU, and is comparable to Gated.

(b) At a fixed sparsity level ($L_0 = 128$), scaling laws are steeper for TopK than ReLU.[3]

Figure 2: Comparison between TopK and other activation functions.

and the decoder is unchanged. The training loss is simply $\mathcal{L} = ||x - \hat{x}||_2^2$.

Using $k$-sparse autoencoders has a number of benefits:

- It removes the need for the $L_1$ penalty. $L_1$ is an imperfect approximation of $L_0$, and it introduces a bias of shrinking all positive activations toward zero (subsection 5.1).

- It enables setting the $L_0$ directly, as opposed to tuning an $L_1$ coefficient $\lambda$, enabling simpler model comparison and rapid iteration. It can also be used in combination with arbitrary activation functions.[2]

- It empirically outperforms baseline ReLU autoencoders on the sparsity-reconstruction frontier (Figure 2a), and this gap increases with scale (Figure 2b).

- It increases monosemanticity of random activating examples by effectively clamping small activations to zero (subsection 4.3).

### 2.4 PREVENTING DEAD LATENTS

Dead latents pose another significant difficulty in autoencoder training. In larger autoencoders, an increasingly large proportion of latents stop activating entirely at some point in training. For example, Templeton et al. (2024) train a 34 million latent autoencoder with only 12 million alive latents, and in our ablations we find up to 90% dead latents[4] when no mitigations are applied (Figure 13). This results in substantially worse MSE and makes training computationally wasteful. We find two important ingredients for preventing dead latents: we initialize the encoder to the transpose of the decoder,[5] and we use an auxiliary loss that models reconstruction error using the top-$k_{\mathrm{aux}}$ dead latents (see subsection A.2 for more details). Using these techniques, even in our largest (16 million latent) autoencoder only 7% of latents are dead.

## 3 SCALING LAWS

### 3.1 NUMBER OF LATENTS

Due to the broad capabilities of frontier models such as GPT-4, we hypothesize that faithfully representing model state will require large numbers of sparse features. We consider two primary approaches to choose autoencoder size and token budget:

---

[2]In our code, we also apply a ReLU to guarantee activations to be non-negative. However, the training curves are indistinguishable, as the $k$ largest activations are almost always positive for reasonable choices of $k$.

[3]Non-TopK cannot set a precise $L_0$, so we interpolated using a piecewise linear function in log-log space.

[4]We follow Templeton et al. (2024), and consider a latent dead if it has not activated in 10 million tokens

[5]This strategy is also presented in concurrent work (Conerly et al., 2024).

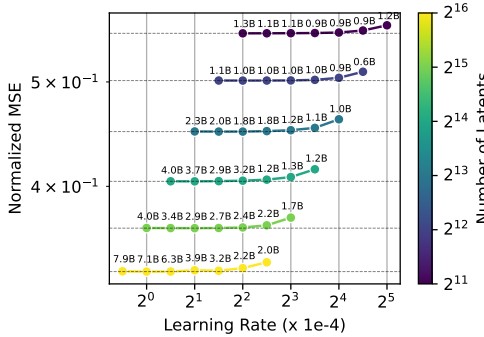 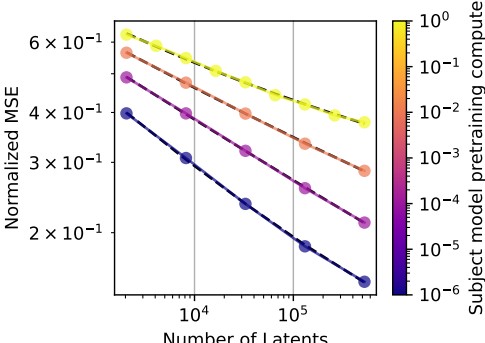

Figure 3: Varying the learning rate jointly with the number of latents. Number of tokens to convergence shown above each point.

Figure 4: Larger subject models in the GPT-4 family require more latents to get to the same MSE ($k = 32$).

### 3.1.1 TRAINING TO COMPUTE-MSE FRONTIER ($L(C)$)

Firstly, following Lindsey et al. (2024), we train autoencoders to the optimal MSE given the available compute, disregarding convergence. This method was introduced for pre-training language models (Kaplan et al., 2020; Hoffmann et al., 2022). We find that MSE follows a power law $L(C)$ of compute, though the smallest models are off trend (Figure 1).

However, latents are the important artifact of training (not reconstruction predictions), whereas for language models we typically care only about token predictions. Comparing MSE across different $n$ is thus not a fair comparison — the latents have a looser information bottleneck with larger $n$, so lower MSE is more easily achieved. Thus, this approach is arguably unprincipled for autoencoder training.

### 3.1.2 TRAINING TO CONVERGENCE ($L(N)$)

We also look at training autoencoders to convergence (within some $\epsilon$). This gives a bound on the best possible reconstruction achievable by our training method if we disregard compute efficiency. In practice, we would ideally train to some intermediate token budget between $L(N)$ and $L(C)$.

We find that the largest learning rate that converges scales with $1/\sqrt{n}$ (Figure 3). We also find that the optimal learning rate for $L(N)$ is about four times smaller than the optimal learning rate for $L(C)$.

We find that the number of tokens to convergence increases as approximately $\Theta(n^{0.6})$ for GPT-2 small and $\Theta(n^{0.65})$ for GPT-4 (Figure 9). This must break at some point – if token budget continues to increase sublinearly, the number of tokens each latent receives gradient signal on would approach zero.[6]

### 3.1.3 IRREDUCIBLE LOSS

Scaling laws sometimes include an irreducible loss term $e$, such that $y = \alpha x^\beta + e$ (Henighan et al., 2020). We find that including an irreducible loss term substantially improves the quality of our fits for both $L(C)$ and $L(N)$.

It was initially not clear to us that there should be a nonzero irreducible loss. One possibility is that there are other kinds of structures in the activations. In the extreme case, unstructured noise in the activations is substantially harder to model and would have an exponent close to zero (Appendix G). Existence of some unstructured noise would explain a bend in the power law.

---

[6]One slight complication is that in the infinite width limit, TopK autoencoders with our initialization scheme are actually optimal at init using our init scheme (subsection A.1), so this allows for an exponent very slightly less than 1; however, this happens very slowly with $n$ so is unlikely to be a major factor at realistic scales.

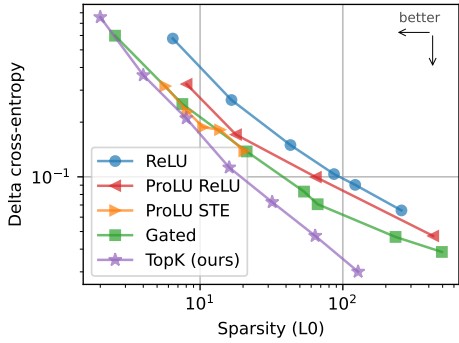 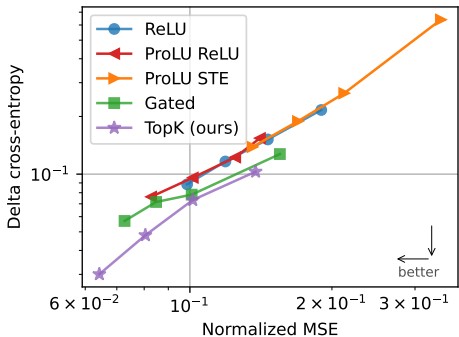

(a) For a fixed number of latents ($n = 2^{17} = 131072$), the downstream-loss/sparsity trade-off is better for TopK autoencoders than for other activation functions.

(b) For a fixed sparsity level ($L_0 = 128$), a given MSE level leads to a lower downstream-loss for TopK autoencoders than for other activation functions.

Figure 5: Comparison between TopK and other activation functions on downstream loss. Comparisons done for GPT-2 small, see Figure 11 for GPT-4.

#### 3.1.4 JOINTLY FITTING SPARSITY ($L(N, K)$)

We find that MSE follows a joint scaling law along the number of latents $n$ and the sparsity level $k$ (Figure 1b). Because reconstruction becomes trivial as $k$ approaches $d_{model}$, this scaling law only holds for the small $k$ regime. Our joint scaling law fit on GPT-4 autoencoders is:

$$L(n, k) = \exp(\alpha + \beta_k \log(k) + \beta_n \log(n) + \gamma \log(k) \log(n)) + \exp(\zeta + \eta \log(k)) \quad (3)$$

with $\alpha = -0.50$, $\beta_k = 0.26$, $\beta_n = -0.017$, $\gamma = -0.042$, $\zeta = -1.32$, and $\eta = -0.085$. We can see that $\gamma$ is negative, which means that the scaling law $L(N)$ gets steeper as $k$ increases. $\eta$ is negative too, which means that the irreducible loss decreases with $k$.

### 3.2 SUBJECT MODEL SIZE $L_s(N)$

Since language models are likely to keep growing in size, we would also like to understand how sparse autoencoders scale with the subject model. We find that if we hold $k$ constant, larger subject models require larger autoencoders to achieve the same MSE, and the exponent is worse (Figure 4).

## 4 EVALUATION

We demonstrated in section 3 that our larger autoencoders scale well in terms of MSE and sparsity (see also a comparison of activation functions in subsection 5.2). However, the end goal of autoencoders is not to improve the sparsity-reconstruction frontier (which degenerates in the limit[7]), but rather to find features useful for applications, such as mechanistic interpretability. Therefore, we measure autoencoder quality with the following metrics:

1. **Downstream loss**: How good is the language model loss if the residual stream latent is replaced with the autoencoder reconstruction of that latent? (subsection 4.1)

2. **Probe loss**: Do autoencoders recover features that we believe they might have? (subsection 4.2)

3. **Explainability**: Are there simple explanations that are both necessary and sufficient for the activation of the autoencoder latent? (subsection 4.3)

4. **Ablation sparsity**: Does ablating individual latents have a sparse effect on downstream logits? (subsection 4.5)

---

[7]Improving the reconstruction-sparsity frontier is not always strictly better. An infinitely wide maximally sparse ($k = 1$) autoencoder can perfectly reconstruct by assigning latents densely in $\mathbb{R}^d$, while being completely structure-less and uninteresting.

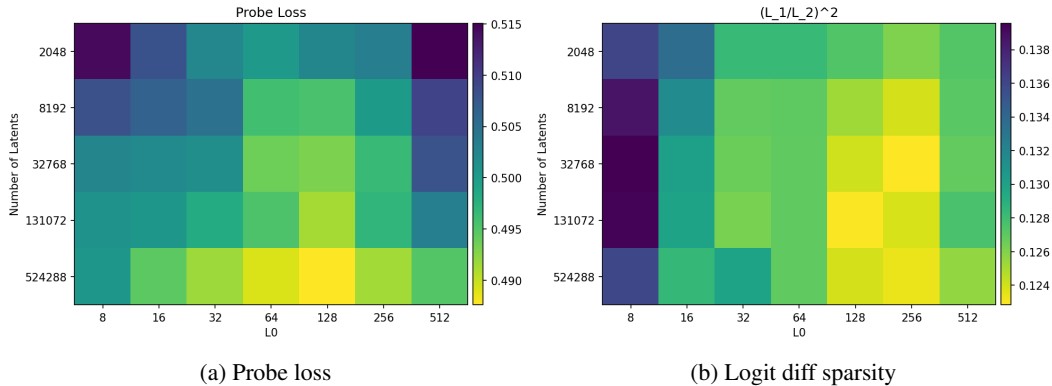

(a) Probe loss          (b) Logit diff sparsity

Figure 6: The probe loss and logit diff metrics as a function of number of total latents $n$ and active latents $k$, for GPT-2 small autoencoders. More total latents (higher $n$) generally improves all metrics (yellow = better). Both metrics are worse at $L_0 = 512$, a regime in which solutions are dense (see subsection E.5).

These metrics provide evidence that autoencoders generally get better when the number of total latents increases. The impact of the number of active latents $L_0$ is more complicated. Increasing $L_0$ makes explanations based on token patterns worse, but makes probe loss and ablation sparsity better. All of these trends also break when $L_0$ gets close to $d_{model}$, a regime in which latents also become quite dense (see subsection E.5 for detailed discussion).

## 4.1 DOWNSTREAM LOSS

An autoencoder with non-zero reconstruction error may not succeed at modeling the features most relevant for behavior (Braun et al., 2024). To measure whether we model features relevant to language modeling, we follow prior work (Bills et al., 2023; Cunningham et al., 2023; Bricken et al., 2023; Braun et al., 2024) and consider downstream Kullback-Leibler (KL) divergence and cross-entropy loss.[8] In both cases, we test an autoencoder by replacing the residual stream by the reconstructed value during the forward pass, and seeing how it affects downstream predictions. We find that $k$-sparse autoencoders improve more on downstream loss than on MSE over prior methods (Figure 5a). We also find that MSE has a clean power law relationship with both KL divergence, and difference of cross entropy loss (Figure 5b), when keeping sparsity $L_0$ fixed and only varying autoencoder size.

One additional issue is that raw loss numbers alone are difficult to interpret—we would like to know how good it is in an absolute sense. Prior work (Bricken et al., 2023; Rajamanoharan et al., 2024) use the loss of ablating activations to zero as a baseline and report the fraction of loss recovered from that baseline. However, because ablating the residual stream to zero causes very high downstream loss, this means that even very poorly explaining the behavior can result in high scores.[9]

Instead, we believe a more natural metric is to consider the relative amount of pretraining compute needed to train a language model of comparable downstream loss. For example, when our 16 million latent autoencoder is substituted into GPT-4, we get a language modeling loss corresponding to 10% of the pretraining compute of GPT-4.

## 4.2 RECOVERING KNOWN FEATURES WITH 1D PROBES

If we expect that a specific feature (e.g sentiment, language identification) should be discovered by a high quality autoencoder, then one metric of autoencoder quality is to check whether these features are present. Based on this intuition, we curated a set of 61 binary classification datasets (details in

---

[8]Because a perfect reconstruction would lead to a non-zero cross-entropy loss, we actually consider the difference to the perfect-autoencoder cross-entropy ("delta cross-entropy").

[9]For completeness, the zero-ablation fidelity metric of our 16M autoencoder is 98.2%.

Table 1). For each task, we train a 1d logistic probe on each latent using the Newton-Raphson method to predict the task, and record the best cross entropy loss (across latents).[10] That is:

$$\min_{i,w,b} \mathbb{E}\left[-y \log \sigma\left(wz_i + b\right) - (1 - y) \log\left(1 - \sigma\left(wz_i + b\right)\right)\right] \tag{4}$$

where $z_i$ is the $i$th pre-activation latent, and $y$ is a binary label. We average the loss across tasks.

Results on GPT-2 small are shown in Figure 6a. We find that probe score increases and then decreases as $k$ increases. We find that TopK generally achieves better probe scores than ReLU (Figure 23), and both are substantially better than when using directly residual stream channels. See Figure 32 for results on several GPT-4 autoencoders: we observe that this metric improves throughout training, despite there being no supervised training signal; and we find that it beats a baseline using channels of the residual stream. See Figure 33 for scores broken down by component.

This metric has the advantage that it is computationally cheap. However, it also has a major limitation, which is that it leans on strong assumptions about what kinds of features are natural.

### 4.3 FINDING SIMPLE EXPLANATIONS FOR FEATURES

Anecdotally, our autoencoders find many features that have quickly recognizable patterns that suggest explanations when viewing random activations (subsection E.1). However, this can create an "illusion" of interpretability (Bolukbasi et al., 2021), where explanations are overly broad, and thus have good recall but poor precision. For example, Bills et al. (2023) propose an automated interpretability score which disproportionately depends on recall. They find a feature activating at the end of the phrase "don't stop" or "can't stop", but an explanation activating on all instances of "stop" achieves a high interpretability score. As we scale autoencoders and the features get sparser and more specific, this kind of failure becomes more severe.

Unfortunately, precision is extremely expensive to evaluate when the simulations are using GPT-4 as in Bills et al. (2023). As an initial exploration, we focus on an improved version of Neuron to Graph (N2G) (Foote et al., 2023), a substantially less expressive but much cheaper method that outputs explanations in the form of collections of n-grams with wildcards. In the future, we would like to explore ways to make it more tractable to approximate precision for arbitrary English explanations.

To construct a N2G explanation, we start with some sequences that activate the latent. For each one, we find the shortest suffix that still activates the latent.[11] We then check whether any position in the n-gram can be replaced by a padding token, to insert wildcard tokens. We also check whether the explanation should be dependent on absolute position by checking whether inserting a padding token at the beginning matters. We use a random sample of up to 16 nonzero activations to build the graph, and another 16 as true positives for computing recall.

Results for GPT-2 small are found in Figure 25a and 25b. Note that dense token patterns are trivial to explain, thus $n = 2048, k = 512$ latents are easy to explain on average since many latents activate extremely densely (see subsection E.5)[12]. In general, autoencoders with more total latents and fewer active latents are easiest to model with N2G.

We also obtain evidence that TopK models have fewer spurious positive activations thanReLU models. N2G explanations have significantly better recall (>1.5x) and only slightly worse precision (>0.9x) for TopK models with the same $n$ (resulting in better F1 scores) and similar $L_0$ (Figure 24).

### 4.4 EXPLANATION RECONSTRUCTION

When our goal is for a model's activations to be interpretable, one question we can ask is: how much performance do we sacrifice if we use only the parts of the model that we can interpret?

---

[10] This is similar to the approach of Gurnee et al. (2023), but always using $k = 1$, and regressing on autoencoder latents instead of neurons.

[11] with at least half the original activation strength

[12] This highlights an issue with our precision/recall metrics, which care only about binarized values. We also propose a more expensive metric which uses simulated values subsection 4.4 and addresses this issue.

Our downstream loss metric measures how much of the performance we're capturing (but our features could be uninterpretable), and our explanation based metric measures how monosemantic our features are (but they might not explain most of the model). This suggests combining our downstream loss and explanation metrics, by using our explanations to simulate autoencoder latents, and then checking downstream loss after decoding. This metric also has the advantage that it values both recall and precision in a way that is principled, and also values recall more for latents that activate more densely.

We tried this with N2G explanations. N2G produces a simulated value based on the node in the trie, but we scale this value to minimize variance explained. Specifically, we compute $E[sa]/E[s^2]$, where $s$ is the simulated value and $a$ is the true value, and we estimate this quantity over a training set of tokens. Results for GPT-2 are shown in Figure 7.

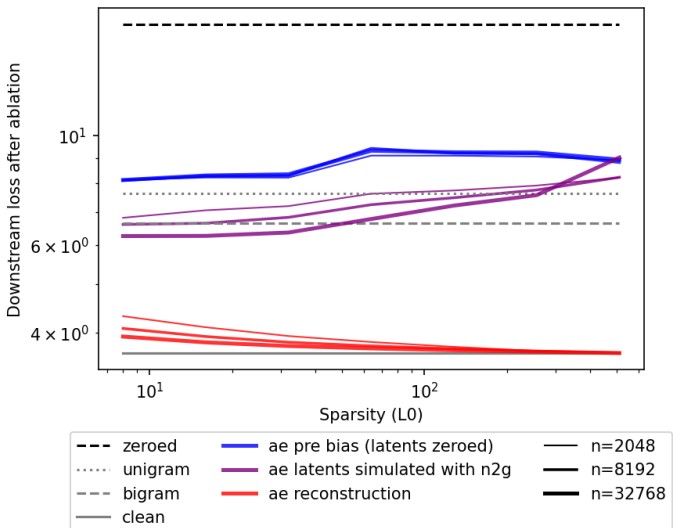

Figure 7: Downstream loss on GPT-2 with various residual stream interventions at layer 8. When every latent in the autoencoder is explained with N2G and the model is run using the N2G simulations instead of the real latent activations (purple), we find that larger and sparser autoencoders have better downstream loss. In the best case, running the model with N2G simulated latents performs better than a baseline of bigram language modeling, showing some nontrivial degree of explanation. However, most of the original autoencoder's reconstruction (red) remains unexplained by N2G explanations.

## 4.5 SPARSITY OF ABLATION EFFECTS

If the underlying computations learned by a language model are sparse, one hypothesis is that natural features are not only sparse in terms of activations, but also in terms of downstream effects (Olah et al., 2024). Anecdotally, we observed that ablation effects often are interpretable (see our visualizer). Therefore, we developed a metric to measure the sparsity of downstream effects on the output logits.

At a particular token index, we obtain the latents at the residual stream, and proceed to ablate each autoencoder latent one by one, and compare the resulting logits before and after ablation. This process leads to $V$ logit differences per ablation and affected token, where $V$ is the size of the token vocabulary. Because a constant difference at every logit does not affect the post-softmax probabilities, we subtract at each token the median logit difference value. Finally, we concatenate these vectors together across some set of $T$ future tokens (at the ablated index or later) to obtain a vector of $V \cdot T$ total numbers. We then measure the sparsity of this vector via $(\frac{L_1}{L_2})^2$, which corresponds to an "effective number of vocab tokens affected". We normalize by $V \cdot T$ to have a fraction between 0 and 1, with smaller values corresponding to sparser effects.

We perform this for various autoencoders trained on the post-MLP residual stream at layer 8 in GPT-2 small, with $T = 16$. Results are shown in Figure 6b. Promisingly, models trained with larger $k$ have latents with sparser effects. However, the trend reverses at $k = 512$, indicating that as $k$ approaches $d_{\text{model}} = 768$, the autoencoder learns latents with less interpretable effects. Note that latents are

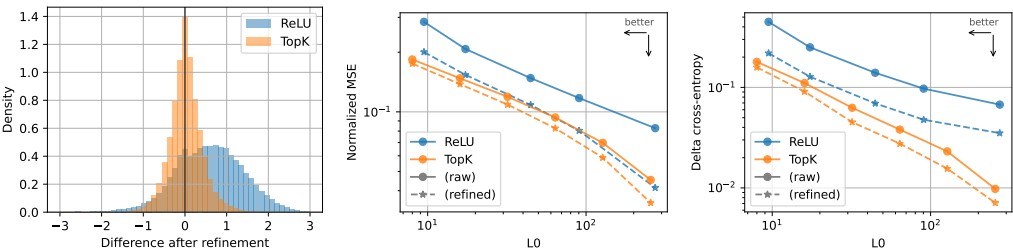

Figure 8: Latent activations can be refined to improve reconstruction from a frozen set of latents. For ReLU autoencoders, the refinement is biased toward positive values, consistent with compensating for the shrinkage caused by the $L_1$ penalty. For TopK autoencoders, the refinement is not biased, and also smaller in magnitude. The refinement only closes part of the gap between ReLU and TopK.

sparse in an absolute sense, having a $(\frac{L_1}{L_2})^2$ of 10-14% , whereas ablating residual stream channels gives 60% (slightly better than the theoretical value of $\sim \frac{2}{\pi}$ for random vectors).

## 5 UNDERSTANDING THE TOPK ACTIVATION FUNCTION

### 5.1 TOPK PREVENTS ACTIVATION SHRINKAGE

A major drawback of the $L_1$ penalty is that it tends to shrink all activations toward zero (Tibshirani, 1996). Our proposed TopK activation function prevents activation shrinkage, as it entirely removes the need for an $L_1$ penalty. To empirically measure the magnitude of activation shrinkage, we consider whether different (and potentially larger) activations would result in better reconstruction given a fixed decoder. We first run the encoder to obtain a set of activated latents, save the sparsity mask, and then optimize only the nonzero values to minimize MSE.[13] This refinement method has been proposed multiple times such as in $k$-SVD (Aharon et al., 2006), the relaxed Lasso (Meinshausen, 2007), or ITI (Maleki, 2009). We solve for the optimal activations with a positivity constraint using projected gradient descent.

This refinement procedure tends to increase activations in ReLU models on average, but not in TopK models (Figure 8a), which indicates that TopK is not impacted by activation shrinkage. The magnitude of the refinement is also smaller for TopK models than for ReLU models. In both ReLU and TopK models, the refinement procedure noticeably improves the reconstruction MSE (Figure 8b), and the downstream next-token-prediction cross-entropy (Figure 8c). However, this refinement only closes part of the gap between ReLU and TopK models.

### 5.2 COMPARISON WITH OTHER ACTIVATION FUNCTIONS

Other recent works on sparse autoencoders have proposed different ways to address the $L_1$ activation shrinkage, and Pareto improve the $L_0$-MSE frontier (Wright & Sharkey, 2024; Taggart, 2024; Rajamanoharan et al., 2024). Wright & Sharkey (2024) propose to fine-tune a scaling parameter per latent, to correct for the $L_1$ activation shrinkage. In Gated sparse autoencoders (Rajamanoharan et al., 2024), the selection of which latents are active is separate from the estimation of the activation magnitudes. This separation allows autoencoders to better estimate the activation magnitude, and avoid the $L_1$ activation shrinkage. Another approach is to replace the ReLU activation function with a ProLU (Taggart, 2024) (also known as TRec (Konda et al., 2014), or JumpReLU (Erichson et al., 2019)), which sets all values below a positive threshold to zero $J_\theta(x) = x \cdot \mathbf{1}_{(x>\theta)}$. Because the parameter $\theta$ is non-differentiable, it requires a approximate gradient such as a ReLU equivalent (ProLU-ReLU) or a straight-through estimator (ProLU-STE) (Taggart, 2024).

We compared these different approaches in terms of reconstruction MSE, number of active latents $L_0$, and downstream cross-entropy loss (Figure 2 and 5). We find that they significantly improve the reconstruction-sparsity Pareto frontier, with TopK having the best performance overall.

---

[13]unlike the "inference-time optimization" procedure in Nanda et al. (2024)

## 6    LIMITATIONS AND FUTURE DIRECTIONS

We believe many improvements can be made to our autoencoders.

- TopK forces every token to use exactly $k$ latents, which is likely suboptimal. Ideally we would constrain $\mathbb{E}[L_0]$ rather than $L_0$.
- The optimization can likely be greatly improved, for example with learning rate scheduling,[14] better optimizers, and better aux losses for preventing dead latents.
- Much more could be done to understand what metrics best track relevance to downstream applications, and to study those applications themselves. Applications include: finding vectors for steering behavior, doing anomaly detection, identifying circuits, and more.
- We're excited about work in the direction of combining MoE (Shazeer et al., 2017) and autoencoders, which would substantially improve the asymptotic cost of autoencoder training, and enable much larger autoencoders.
- A large fraction of the random activations of features we find, especially in GPT-4, are not yet adequately monosemantic. We believe that with improved techniques and greater scale[15] this is potentially surmountable.
- Our probe based metric is quite noisy, which could be improved by having a greater breadth of tasks and higher quality tasks.
- While we use n2g for its computational efficiency, it is only able to capture very simple patterns. We believe there is a lot of room for improvement in terms of more expressive explanation methods that are also cheap enough to simulate to estimate explanation precision.
- A context length of 64 tokens is potentially too few tokens to exhibit the most interesting behaviors of GPT-4.

## 7    RELATED WORK

Sparse coding on an over-complete dictionary was introduced by Mallat & Zhang (1993). Olshausen & Field (1996) refined the idea by proposing to learn the dictionary from the data, without supervision. This approach has been particularly influential in image processing, as seen for example in (Mairal et al., 2014). Later, Hinton & Salakhutdinov (2006) proposed the autoencoder architecture to perform dimensionality reduction. Combining these concepts, sparse autoencoders were developed (Lee et al., 2007; Le et al., 2013; Konda et al., 2014) to train autoencoders with sparsity priors, such as the $L_1$ penalty, to extract sparse features. Makhzani & Frey (2013) refined this concept by introducing $k$-sparse autoencoders, which use a TopK activation function instead of the $L_1$ penalty. Makelov et al. (2024) evaluates autoencoders using a metric that measures recovery of features from previously discovered circuits.

More recently, sparse autoencoders were applied to language models (Yun et al., 2021; Lee Sharkey, 2022; Bricken et al., 2023; Cunningham et al., 2023), and multiple sparse autoencoders were trained on small open-source language models (Marks, 2023; Bloom, 2024; Mossing et al., 2024). Marks et al. (2024) showed that the resulting features from sparse autoencoders can find sparse circuits in language models. Wright & Sharkey (2024) pointed out that sparse autoencoders are subject to activation shrinking from $L_1$ penalties, a property of $L_1$ penalties first described in Tibshirani (1996). Taggart (2024) and Rajamanoharan et al. (2024) proposed to use different activation functions to address activation shrinkage in sparse autoencoders. Braun et al. (2024) proposed to train sparse autoencoders on downstream KL instead of reconstruction MSE.

Kaplan et al. (2020) studied scaling laws for language models which examine how loss varies with various hyperparameters. Clark et al. (2022) explore scaling laws related to sparsity using a bilinear fit. Lindsey et al. (2024) studied scaling laws specifically for autoencoders, defining the loss as a specific balance of reconstruction and sparsity (rather than simply reconstruction, while holding sparsity fixed).

---

[14]Anecdotally, we also found that lowering learning rates helped with decreasing dead latents.

[15]both in number of latents and in training tokens

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

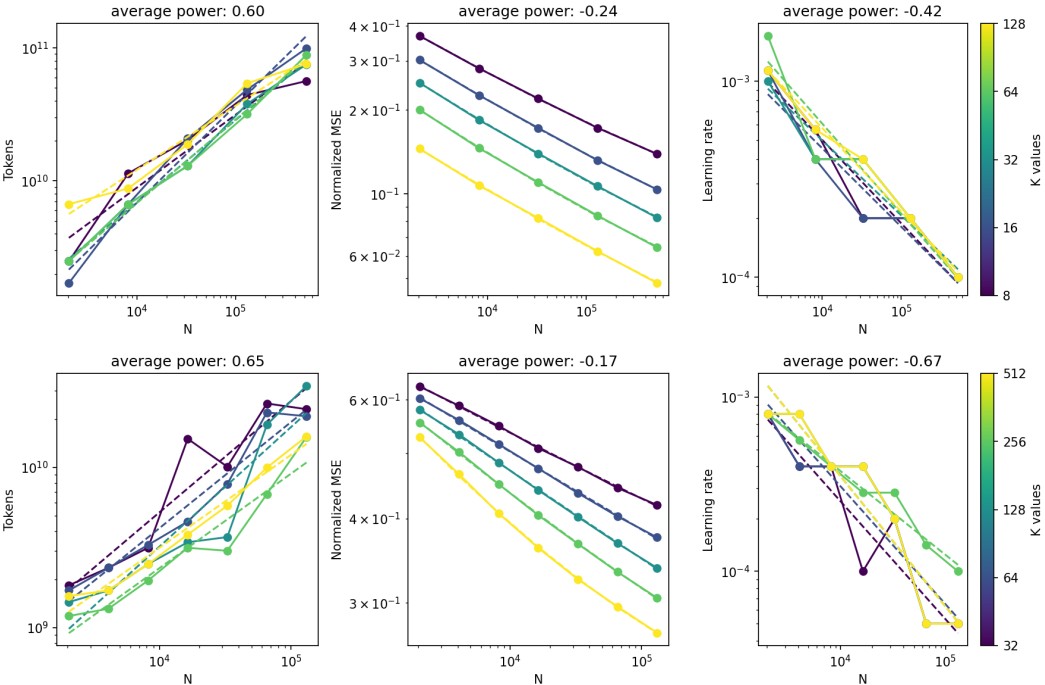

Figure 9: Token budget, MSE, and learning rate power laws, averaged across values of $k$. First row is GPT-2, second row is GPT-4. Note that individual fits are noisy (especially for GPT-4) since learning rate sweeps are coarse, and token budget depends on learning rate.

Zeyu Yun, Yubei Chen, Bruno A Olshausen, and Yann LeCun. Transformer visualization via dictionary learning: contextualized embedding as a linear superposition of transformer factors. *arXiv preprint arXiv:2103.15949*, 2021.

Yanli Zhao, Andrew Gu, Rohan Varma, Liang Luo, Chien-Chin Huang, Min Xu, Less Wright, Hamid Shojanazeri, Myle Ott, Sam Shleifer, et al. Pytorch fsdp: experiences on scaling fully sharded data parallel. *arXiv preprint arXiv:2304.11277*, 2023.

Ben Zhou, Daniel Khashabi, Qiang Ning, and Dan Roth. "going on a vacation" takes longer than "going for a walk": A study of temporal commonsense understanding. In *Proceedings of the 2019 Conference on Empirical Methods in Natural Language Processing and the 9th International Joint Conference on Natural Language Processing (EMNLP-IJCNLP)*, 2019. URL https://arxiv.org/abs/1909.03065.

# A   OPTIMIZATION

## A.1   INITIALIZATION

We initialize our autoencoders as follows:

- We initialize the bias $b_{pre}$ to be the geometric median of a sample set of data points, following Bricken et al. (2023).
- We initialize the encoder directions parallel to the respective decoder directions, so that the corresponding latent read/write directions are the same[16] Directions are chosen uniformly randomly.

---

[16]This is done only at initialization; we do not tie the parameters as in Cunningham et al. (2023). This strategy is also presented in concurrent work (Conerly et al., 2024).

- We scale decoder latent directions to be unit norm at initialization (and also after each training step), following Bricken et al. (2023).

- For baseline models we use torch default initialization for encoder magnitudes. For TopK models, we initialized the magnitude of the encoder such that the magnitude of reconstructed vectors match that of the inputs. However, in our ablations we find this has no effect or a weak negative effect (Figure 14).[17]

## A.2 AUXILIARY LOSS

We define an auxiliary loss (AuxK) similar to "ghost grads" (Jermyn & Templeton, 2024) that models the reconstruction error using the top-$k_{\text{aux}}$ dead latents (typically $k_{\text{aux}} = 512$). Latents are flagged as dead during training if they have not activated for some predetermined number of tokens (typically 10 million). Then, given the reconstruction error of the main model $e = x - \hat{x}$, we define the auxiliary loss $\mathcal{L}_{\text{aux}} = ||e - \hat{e}||_2^2$, where $\hat{e} = W_{\text{dec}} z$ is the reconstruction using the top-$k_{\text{aux}}$ dead latents. The full loss is then defined as $\mathcal{L} + \alpha \mathcal{L}_{\text{aux}}$, where $\alpha$ is a small coefficient (typically $1/32$). Because the encoder forward pass can be shared (and dominates decoder cost and encoder backwards cost, see Appendix D), adding this auxiliary loss only increases the computational cost by about 10%.

We found that the AuxK loss very occasionally NaNs at large scale, and zero it when it is NaN to prevent training run collapse.

## A.3 OPTIMIZER

We use the Adam optimizer (Kingma & Ba, 2014) with $\beta_1 = 0.9$ and $\beta_2 = 0.999$, and a constant learning rate. We tried several learning rate decay schedules but did not find consistent improvements in token budget to convergence. We also did not find major benefits from tuning $\beta_1$ and $\beta_2$.

We project away gradient information parallel to the decoder vectors, to account for interaction between Adam and decoder normalization, as described in Bricken et al. (2023).

### A.3.1 ADAM EPSILON

By convention, we average the gradient across the batch dimension. As a result, the root mean square (RMS) of the gradient can often be very small, causing Adam to no longer be loss scale invariant. We find that by setting epsilon sufficiently small, these issues are prevented, and that $\varepsilon$ is otherwise not very sensitive and does not result in significant benefit to tune further. We use $\varepsilon = 6.25 \times 10^{-10}$ in many experiments in this paper, though we reduced it further for some of the largest runs to be safe.

### A.3.2 GRADIENT CLIPPING

When scaling the GPT-4 autoencoders, we found that gradient clipping was necessary to prevent instability and divergence at higher learning rates. We found that gradient clipping substantially affected $L(C)$ but not $L(N)$. We did not use gradient clipping for the GPT-2 small runs.

## A.4 BATCH SIZE

Larger batch sizes are critical for allowing much greater parallelism. Prior work tends to use batch sizes like 2048 or 4096 tokens (Bricken et al., 2023; Conerly et al., 2024; Rajamanoharan et al., 2024). To gain the benefits of parallelism, we use a batch size of 131,072 tokens for most of our experiments.

While batch size affects $L(C)$ substantially, we find that the $L(N)$ loss does not depend strongly on batch size when optimization hyperparameters are set appropriately (Figure 10).

---

[17]Note that the scaling factor has nontrivial interaction with $n$, and scales between $\Theta(1/\sqrt{k})$ and $\Theta(1/k)$. This scheme has the advantage that is optimal at init in the infinite width limit. We did not try simpler schemes like scaling by $\Theta(1/\sqrt{k})$.

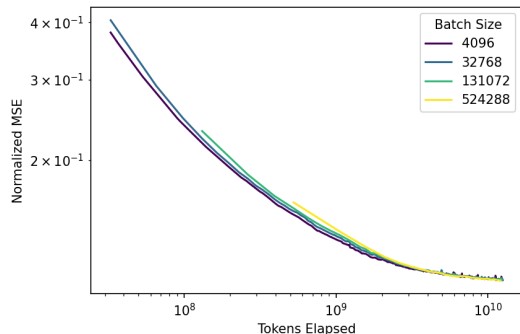

Figure 10: With correct hyperparameter settings, different batch sizes converge to the same $L(N)$ loss (gpt2small).

### A.5    WEIGHT AVERAGING

We find that keeping an exponential moving average (EMA) Ruppert (1988) of the weights slightly reduces sensitivity to learning rate by allowing slightly higher learning rates to be tolerated. Due to its low cost, we use EMA in all experiments. We use an EMA coefficient of 0.999, and did not find a substantial benefit to tuning it.

We use a bias-correction similar to that used in Kingma & Ba (2014). Despite this, the early steps of EMA are still generally worse than the original model. Thus for the $L(C)$ experiments, we take the min of the EMA model's and non-averaged model's validation losses.

### A.6    OTHER DETAILS

- For the main MSE loss, we compute an MSE normalization constant once at the beginning of training, and do not do any loss normalization per batch.
- For the AuxK MSE loss, we compute the normalization per token, because the scale of the error changes throughout training.
- In theory, the $b_{pre}$ lr should be scaled linearly with the norm of the data to make the autoencoder completely invariant to input scale. In practice, we find it to tolerate an extremely wide range of values with little impact on quality.
- Anecdotally, we noticed that when decaying the learning rate of an autoencoder previously training at the L(C) loss, the number of dead latents would decrease.

## B    OTHER TRAINING DETAILS

Unless otherwise noted, autoencoders were trained on the residual activation directly after the layernorm (with layernorm weights folded into the attention weights), since this corresponds to how residual stream activations are used. This also causes importance of input vectors to be uniform, rather than weighted by norm[18].

### B.1    TOPK TRAINING DETAILS

We select $k_{aux}$ as a power of two close to $\frac{d_{model}}{2}$ (e.g. 512 for GPT-2 small). We typically select $\alpha = 1/32$. We find that the training is generally not extremely sensitive to the choice of these hyperparameters.

We find empirically that using AuxK eliminates almost all dead latents by the end of training.

Unfortunately, because of compute constraints, we were unable to train our 16M latent autoencoder to $L(N)$, which made it not possible to include the 16M as part of a consistent $L(N)$ series.

---

[18]We believe one of the main impacts of normalizing is that it the first token positions are downweighted in importance (see subsection F.2 for more discussion).

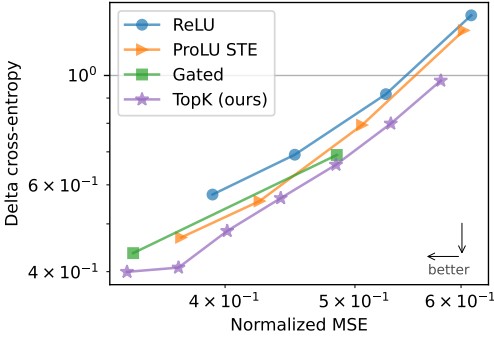

Figure 11: For a fixed sparsity level ($L_0 = 128$), a given MSE leads to a lower downstream-loss for TopK than for other activations functions. (GPT-4). We are less confident about these runs than the corresponding ones for GPT-2, yet the results are consistent (see Figure 5).

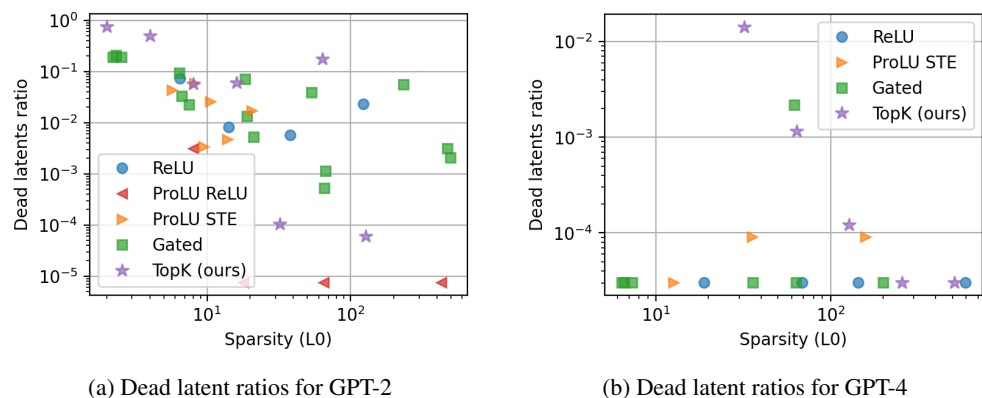

(a) Dead latent ratios for GPT-2

(b) Dead latent ratios for GPT-4

Figure 12: Our baselines generally have few dead latents, similar or less than our TopK models.

## B.2 BASELINE HYPERPARAMETERS

Baseline ReLU autoencoders were trained on GPT-2 small, layer 8. We sweep learning rate in [5e-5, 1e-4, 2e-4, 4e-4], $L_1$ coefficient in [1.7e-3, 3.1e-3, 5e-3, 1e-2, 1.7e-2] and train for 8 epochs of 6.4 billion tokens at a batch size of 131072. We try different resampling periods in [12.5k, 25k] steps, and choose to resample 4 times throughout training. We consider a feature dead if it does not activate for 10 million tokens.

For Gated SAE (Rajamanoharan et al., 2024), we sweep $L_1$ coefficient in [1e-3, 2.5e-3, 5e-3, 1e-2, 2e-2], learning rate in [2.5e-5, 5e-5, 1e-4], train for 6 epochs of 6.4 billion tokens at a batch size of 131072. We resample 4 times throughout training.

For ProLU autoencoders (Taggart, 2024), we sweep $L_1$ coefficient in [5e-4, 1e-3, 2.5e-3, 5e-3, 1e-2, 2e-2], learning rate in [2.5e-5, 5e-5, 1e-4], train for 6 epochs of 6.4 billion tokens at a batch size of 131072. We resample 4 times throughout training. For the ProLU gradient, we try both ProLU-STE and ProLU-ReLU. Note that, consistent with the original work, ProLU-STE autoencoders all have $L_0 < 25$, even for small $L_1$ coefficients.

We used similar settings and sweeps for autoencoders trained on GPT-4. Differences include: replacing the resampling of dead latents with a $L_1$ coefficient warm-up over 5% of training (Conerly et al., 2024); removing the decoder unit-norm constraint and adding the decoder norm in the $L_1$ penalty (Conerly et al., 2024).

Our baselines generally have few dead latents, similar or less than our TopK models (see Figure 12).

# C   TRAINING ABLATIONS

## C.1   DEAD LATENT PREVENTION

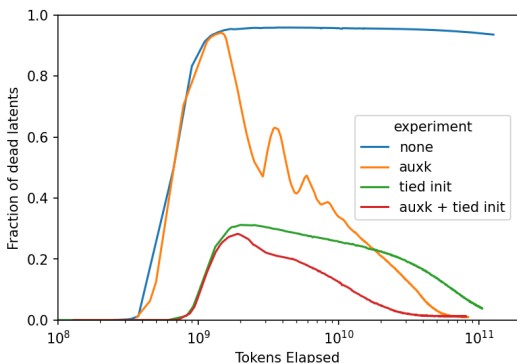

Figure 13: Methods that reduce the number of dead latents (gpt2sm 2M, k=32). With AuxK and/or tied initialization, number of dead latents generally decreases over the course of training, after an early spike.

We find that the reduction in dead latents is mostly due to a combination of the AuxK loss and the tied initialization scheme.

## C.2   INITIALIZATION

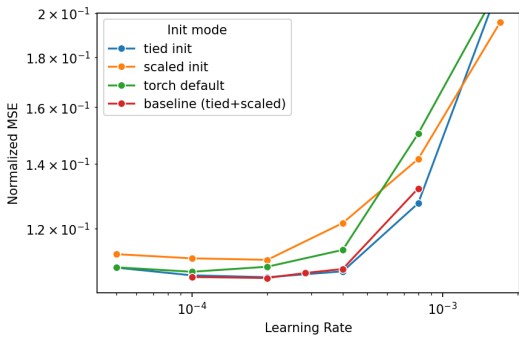

Figure 14: Initialization ablation (gpt2sm 128k, k=32).

We find that tied initialization substantially improves MSE, and that our encoder initialization scheme has no effect when tied initialization is being used, and hurts slightly on its own.

## C.3   $b_{\mathrm{ENC}}$

We find that $b_{\mathrm{enc}}$ does not affect the MSE at convergence. With $b_{\mathrm{enc}}$ removed, the autencoder is equivalent to a JumpReLU where the threshold is dynamically chosen per example such that exactly $k$ latents are active. However, convergence is slightly slower without $b_{\mathrm{enc}}$. We believe this may be confounded by encoder learning rate but did not investigate this further.

## C.4   DECODER NORMALIZATION

After each step we renormalize columns of the decoder to be unit-norm, following Bricken et al. (2023). This normalization (or a modified L1 term, as in Conerly et al. (2024)) is necessary for L1 autoencoders, because otherwise the L1 loss can be gamed by making the latents arbitrarily small. For TopK autoencoders, the normalization is optional. However, we find that it still improves MSE, so we still use it in all of our experiments.

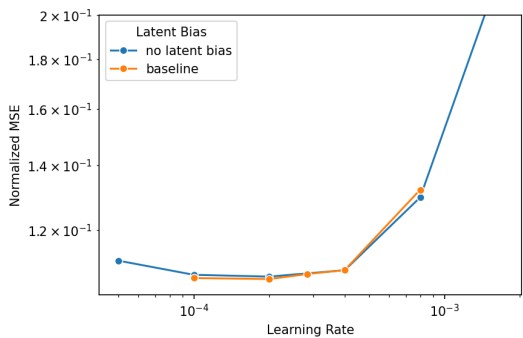

Figure 15: $b_{\text{enc}}$ does not strongly affect loss (gpt2sm 128k, k=32).

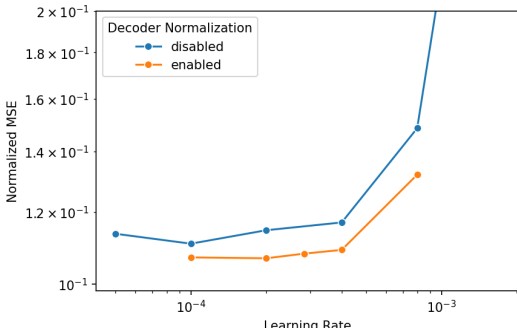

Figure 16: The decoder normalization slightly improves loss (gpt2sm 128k, k=32).

# D SYSTEMS

Scaling autoencoders to the largest scales in this paper would not be feasible without our systems improvements. Model parallelism is necessary once parameters cannot fit on one GPU. A naive implementation can be an order of magnitude slower than our optimized implementation at the very largest scales.

## D.1 PARALLELISM

We use standard data parallel and tensor sharding (Shoeybi et al., 2019), with an additional allgather for the TopK forward pass to determine which $k$ latents should are in the global top k. To minimize the cost of this allgather, we truncate to a capacity factor of 2 per shard—further improvements are possible but would require modifications to NCCL. For the largest (16 million) latent autoencoder, we use 512-way sharding. Large batch sizes (subsection A.4) are very important for reducing the parallelization overhead.

The very small number of layers creates a challenge for parallelism - it makes pipeline parallelism (Huang et al., 2019) and FSDP (Zhao et al., 2023) inapplicable. Additionally, opportunities for communications overlap are limited because of the small number of layers, though we do overlap host to device transfers and encoder data parallel comms for a small improvement.

## D.2 KERNELS

We can take advantage of the extreme sparsity of latents to perform most operations using substantially less compute and memory than naively doing dense matrix multiplication. This was important when scaling to large numbers of latents, both via directly increasing throughput and reducing memory usage.

We use two main kernels:

- `DenseSparseMatmul`: a multiplication between a dense and sparse matrix
- `MatmulAtSparseIndices`: a multiplication of two dense matrices evaluated at a set of sparse indices

Then, we have the following optimizations:

1. The decoder forward pass uses `DenseSparseMatmul`
2. The decoder gradient uses `DenseSparseMatmul`
3. The latent gradient uses `MatmulAtSparseIndices`
4. The encoder gradient uses `DenseSparseMatmul`
5. The pre-bias gradient uses a trick of summing pre-activation gradient across the batch dimension before multiplying with the encoder weights.

Theoretically, this gives a compute efficiency improvement of up to 6x in the limit of sparsity, since the encoder forward pass is the only remaining dense operation. In practice, we indeed find the encoder forward pass is much of the compute, and the pre-activations are much of the memory.

To ensure that reads are coalesced, the decoder weight matrix must also be stored transposed from the typical layout. We also use many other kernels for fusing various operations for reducing memory and memory bandwidth usage.

## E  QUALITATIVE RESULTS

### E.1  SUBJECTIVE LATENT QUALITY

Throughout the project, we stumbled upon many subjectively interesting latents. The majority of latents in our GPT-2 small autoencoders seemed interpretable, even on random positive activations. Furthermore, the ablations typically had predictable effects based on the activation conditions. For example, some features that are potentially part of interesting circuits in GPT-2:

- An unexpected token breaking a repetition pattern (A B C D ... A B X!). This upvotes future pattern breaks (A B Y!), but also upvotes continuation of the pattern right after the break token (A B X $\rightarrow$ D).
- Text within quotes, especially activating when within two nested sets of quotes. Upvotes tokens that close the quotes like " or ', as well as tokens which close multiple sets of quotes at once, such as "' and '".
- Copying/induction of capitalized phrases (A B ... A $\rightarrow$ B)

In GPT-4, we tended to find more complex features, including ones that activate on the same concept in multiple languages, or on complex technical concepts such as algebraic rings.

You can explore for yourself at our viewer.

### E.2  FINDING FEATURES

Typical features are not of particular interest, but having an autoencoder also lets one easily find relevant features. Specifically, one can use gradient-based attribution to quickly compute how relevant latents are to behaviors of interest (Baehrens et al., 2010; Simonyan et al., 2013).

Following the methodologies of Templeton et al. (2024) and Mossing et al. (2024), we found features by using a hand-written prompt, with a set of "positive" and "negative" token predictions, and back-propagating from logit differences to latent values. We then consider latents sorted by activation times gradient value, and inspect the latents.

We tried this methodology with a $n = 2^{17} = 131072$, $k = 32$ GPT-2 autoencoder and were able to quickly find a number of safety relevant latents, such as ones corresponding to profanity or child sexual content. Clamping these latents appeared to have causal effect on the samples. For example, clamping the profanity latent to negative values results in significantly less profanity (some profanity can still be observed in situations where it copies from the prompt).

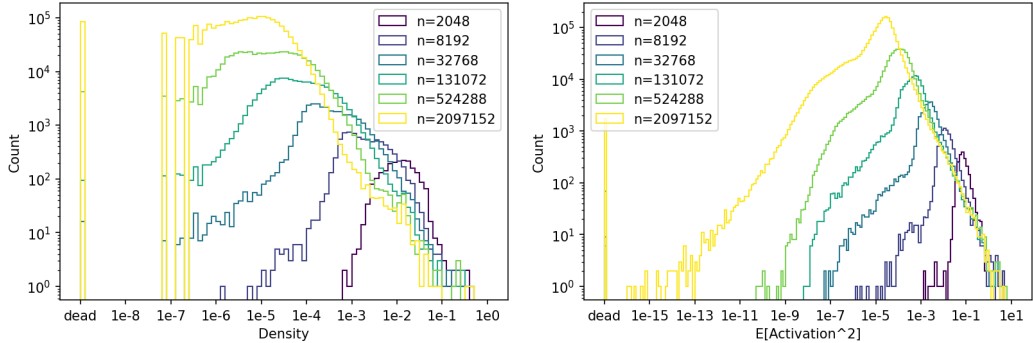

Figure 17: Distributions of latent densities, and average squared activation. Note that we do not observe multiple density modes, as observed in (Bricken et al., 2023). Counts are sampled over 1.5e7 total tokens. Note that because latents that activate every 1e7 tokens are considered dead during training (and thus receive AuxK gradient updates), 1e-7 is in some sense the minimum density, though the AuxK loss term coefficient may allow it to be lower.

### E.3 LATENT ACTIVATION DISTRIBUTIONS

We anecdotally found that latent activation distributions often have multiple modes, especially in early layers.

### E.4 LATENT DENSITY AND IMPORTANCE CURVES

We find that log of latent density is approximately Gaussian (Figure 17). If we define the importance of a feature to be the expected squared activation (an approximation of its marginal impact on MSE), then log importance looks a bit more like a Laplace distribution. Modal density and modal feature importance both decrease with number of total latents (for a fixed $k$), as expected.

### E.5 SOLUTIONS WITH DENSE LATENTS

One option for a sparse autoencoder is to simply learn the $k$ directions that explain the most variance. As $k$ approaches $d_{model}$, this "principal component"-like solution may become competitive with solutions where each latent is used sparsely. In order to check for such solutions, we can simply measure the average density of the $d_{model}$ densest latents. Using this metric, we find that for some hyperparameter settings, GPT-2 small autoencoders find solutions with many dense latents (Figure 18), beginning around $k = 256$ but especially for $k = 512$.

This coincides with when the scaling laws from section 3 begin to break - many of our trends between $k$ and reconstruction loss bend significantly. Also, MSE becomes significantly less sensitive to $n$, at $k = 512$.

### E.6 RECURRING DENSE FEATURES IN GPT-2 SMALL

We manually examined the densest latents across various GPT-2 small layer 8 autoencoders, trained in different ways (e.g. differing numbers of total latents).

The two densest latents are always the same feature: the latent simply activates more and more later in the context ($\sim$ 40% active), and one that activates more and more earlier in the context excluding the first position ($\sim$35% active). Both features look like they want to activate more than they do, with TopK probably preventing it from activating with lower values.

The third densest latent is always a first-token-position feature ($\sim$30% active), which has a modal activation value in a narrow range between 14.6-14.8. Most of its activation values are significantly smaller values, at tokens after the first position; the large value is always at the first token. These smaller values appear uninterpretable; we conjecture these are simply interference with the first

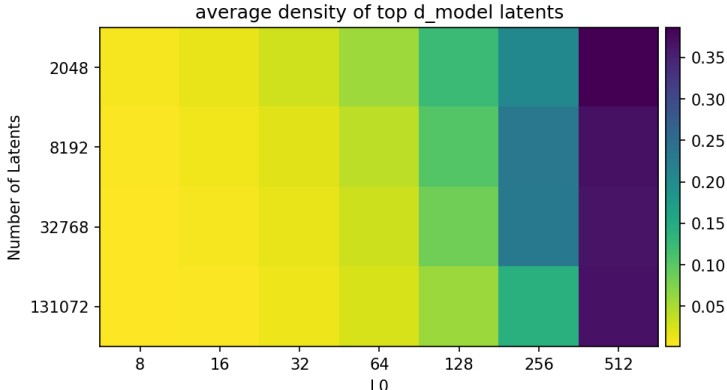

Figure 18: Average density of the $d_{model}$ most-dense features, divided by $L_0$, for different autoencoders. When $k = 512$, the learned autoencoders have many dense features. This corresponds to when ablations stop having sparse effects subsection 4.5, and anecdotally corresponds to noticeably less interpretable features. For $k = 256$, $n = 2048$, there is perhaps an intermediate regime.

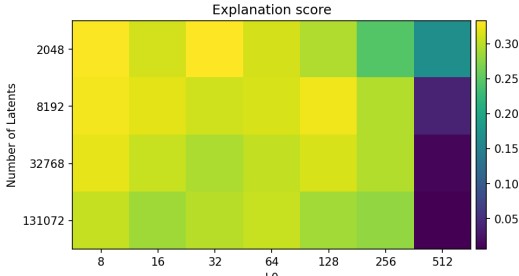

Figure 19: Explanation scores for GPT-2 small autoencoders of different $n$ and $k$, evaluated on 400 randomly chosen latents per autoencoder. It is hard to read off trends, but the explanation score is able to somewhat detect the dense solutions region.

position direction. (Sometimes there are two of these latents, the second with smaller activation values.)

Finally, there is a recurring "repetition" feature that is $\sim$20% dense. Its top activations are mostly highly repetitive sequences, such as series of dates, chapter indices, numbers, punctuations, repeated exact phrases, or other repetitive things such as Chess PGN notation. However, like the first-token-position latents, random activations of this latent are typically appear unrelated and uninterpretable.

Often in the top ten densest latents, we find opposing latents, which have decoder cosine similarity close to $-1$. In particular, the first-token-position feature and the repetition latent both seems to always have an opposite latent. The less dense of the two opposite latents always seems to appear uninterpretable. We conjecture that these are symptoms of optimization failure - the opposite latents cancel out spurious activations in the denser latent.

### E.7 CLUSTERING LATENTS

(Elhage et al., 2022) discuss how underlying features may lie in distinct sub-spaces. If such sub-spaces exists, we hypothesize that the set of latent encoding vectors $W \in \mathbb{R}^{n \times d}$ can be written as a block-diagonal matrix $W' = PWR$, where $P \in \mathbb{R}^{n \times n}$ is a permutation matrix, and $R \in \mathbb{R}^{d \times d}$ is orthogonal. We can then use the singular vector decomposition (SVD) to write $W = U\Sigma V^\top$ and $W' = U'\Sigma'V'^\top$, noting that $U'$ is also block diagonal. Finally, we write $W = P^\top U'\Sigma'V'^\top R^\top = U\Sigma V^\top$, and because the SVD is unique up to a column permutation $P'$, we get $U = P^\top U'P'$. In other words, if $W$ is block-diagonal in some unknown basis, $U$ is also block diagonal up to a permutation of rows and columns.

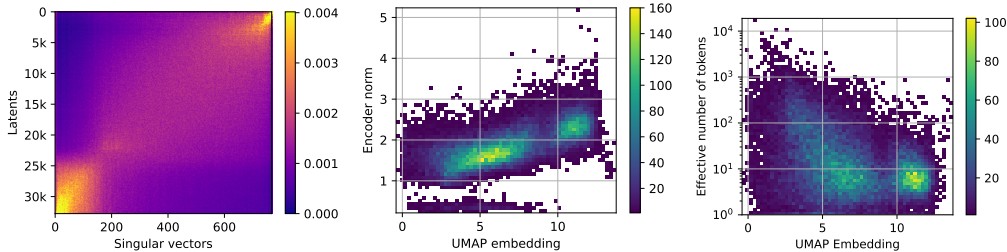

Figure 20: The residual stream seems composed of two separate sub-spaces. About 25% of latents mostly project on a sub-space using 25% of dimensions. These latents tend to have larger encoder norm, and to activate on a smaller number of vocabulary tokens. The remaining 75% of latents mostly project on the remaining 75% of dimensions, and can activate on a larger number of vocabulary tokens.

To find a good permutation of rows, we sorted the rows of $U$ based on how similarly they project on all elements of the singular vector basis. Specifically, we normalized each row to unit norm $\tilde{U}_i = U_i / ||U_i||$ and considered the pairwise euclidean distances $d_{i,j} = ||\tilde{U}_i^2 - \tilde{U}_j^2||$. These pairwise distances were then reduced to a single dimension with a UMAP algorithm (McInnes et al., 2018). The obtained 1-dimensional embedding was then used to order the projections $\tilde{U}_i^2$ (Figure 20a), which reveals two fuzzily separated sub-spaces. These two sub-spaces use respectively about 25% and 75% of the dimensions of the entire vector space.

Interestingly, ordering the columns by singular values is fairly consistent with these two sub-spaces. One reason for this result might by that latents projecting to the first sub-space have different encoder norms than latents projecting to the second sub-space (Figure 20b). This difference in norm can significantly guide the SVD to separate these two sub-spaces.

To further interpret these two sub-spaces, we manually looked at latents from each cluster. We found that latents from the smaller cluster tend to activate on relatively non-diverse vocabulary tokens. To quantify this insight, we first estimated $A_{i,v}$, the average squared activation of latent $i$ on vocabulary token $v$. Then, we normalized the vectors $A_i$ to sum to one, $\tilde{A}_{i,v} = A_{i,v} / \sum_w A_{i,w}$ and computed the effective number of token $m_i = \exp(\sum_v \tilde{A}_{i,v} \log(\tilde{A}_{i,v}))$. The effective number of token is a continuous metric with values in $[1, n_{vocab}]$, and it is equal to $k$ when a latent activates equally on $k$ vocabulary tokens. With this metric, we confirmed quantitatively (Figure 20c) that latents from the smaller cluster all activate on relatively low numbers of vocabulary tokens (less than 100), whereas latents from the larger cluster sometimes activate on a larger numbers of vocabulary tokens (up to 1000).

### E.8 LATENT VISUALIZATIONS

## F MISCELLANEOUS SMALL RESULTS

### F.1 IMPACT OF DIFFERENT LOCATIONS

In a sweep across locations in GPT-2 small, we found that the optimal learning rate varies with layer and location type (MLP delta, attention delta, MLP post, attention post), but was within a factor of two.

Number of tokens needed for convergence is noticeably higher for earlier layers. MSE is lowest at early layers of the residual stream, increasing until the final layer, at which point it drops. Residual stream deltas have MSE peaking around layer 6, with attention delta falling sharply at the final layer.

When ablating to reconstructions, downstream loss and KL get strictly worse with layer. This is despite normalized MSE dropping at late layers. However, there appears to be an exception at final layers (layer 11/12 and especially 12/12 of GPT-2 small), which can have better normalized MSE than earlier layers, but more severe effects on downstream prediction (Figure 27).

**Latent 28719: Adenosine/dopamine receptors**

Top Activating Sequences:

27-g0001)#F1\n\nD2R-like receptors genes generate variants. D2R

aminergic and alpha 2-adrenergic receptors, respectively. Cross-desensitization experiments revealed

oll, [@B117]). Contrasting evidence exist about a role for D4R in reward (Di

]), D1R play a prominent role in the acquisition/maintenance of self-admin

2) agonist (i.e., quinpirole, 0.1 mg

Random Activating Sequences:

of dopamine D2 receptor protein was observed in pial and mes

cur with the notion that an inverted U-shaped relationship in dopamine signaling at the molecular level in the dorsol

ol for the mAChR on intact cells was not affected by pretreatment with IAP

in release of dopamine in the hippocampus where it enhances long-term potentiation

caudate-putamen. The basal extracellular concentration of dopamine in the dorsolateral prefront

**Latent 50233: Comparisons**

Top Activating Sequences:

your hand and wrist.\n\nThe cuff itself is longer than what you may be used to seeing. This is

revised base to be .155 or.063 thicker than had been agreed upon and requested in Exhibit 7

the side\nbutton and make it on the face like the iphone – not only does it currently\nnot

not a solid beige colour where the eyelets sit like it is

.\n\n550 and the springer. Because the axle on a springer is about 2 further forward of

Random Activating Sequences:

the filament. There is a comparably sized channel in the filaments as deduced from electron micrographs

GeV, where it is more than four times greater than for $m_t = 150$GeV, then

of the resonator has, in a manner different from the leg in FIG. 1, been bent

wing narrower than on the upperside, the chocolate-coloured

R$^{+s}$equenceandcysteinesite $hCXCL8.Inthisstudy, we$

**Latent 17441: Ratifications (multilingual)**

Top Activating Sequences:

Arizona ratified the\nproposed sixteenth amendment to the Constitution

rdiggør en opfordring til ikke at ratificere traktaten, er pure opsp

students with a hatred of America.\n\nWith the ratification of H. R. 3077, any

ratification of the jury chosen. Id. This was error

ratified the application of the payment of the money borrowed. This

Random Activating Sequences:

not to ratify its bilateral trade agreement with Vietnam, linking this action

to the bank or to the insurer, impliedly ratified Dorothy's acts. As between Dorothy and Harford

issue of ratification favors neither party, we\nultimately hold that

les États membres et les pays candidats ratifient en premier lieu la convention du Conseil

effective economic boycott of states whose legislatures did not ratify the Equal Rights Amendment to the United States Constitution

Figure 21: Cherry-picked qualitative examples of latents and their activations on top-activating and random-activating sequences

**Latent 63607**

Top Activating Sequences:

Academy of Fine Arts. Her work has been exhibited at Gallery Nik

kowe Akademii Podlaskiej. Administracja i Zarządz

was a founding member of the Academy of Arles in 1668, and the Duke of

in Little Rock, now known as the Academy at Riverdale.\n\nWealth\nRockefeller served as chief

the American Academy of Poets, one of the most prestigious honors in American

Random Activating Sequences:

Mathematics and Computer Technology*, Akad. Nauk. Ukrain. SSR Inst. Mat., Pre

from the Academy of Neuroeconomics and Neuromanagement at Ningbo University

).\n\nPham Mau Quan, Archive for Rational Mechanics and Analysis [**1**],

programs of a museum which he named the Hall of Flame. Mr. Getz began to collect fire apparatus

provided as an educational service of the American Academy of Neurology. It is not a substitute for professional medical

**Latent 6548**

Top Activating Sequences:

see the same preconditioner\n // flag to reuse the preconditioner from the forward

it and disable various parts of below code\n // find out which part of the code creates the

, then all ways to build joins\n * of three items (from two-item joins and

some casting \n cell.backgroundColor = .black\n return cell\n

or lookUpLevel \n /// is Invalid.\n /// Returns AccessDenied if

Random Activating Sequences:

server name in\n 'UserLogin.dbo.aspnet$_Users'$

\n @param This - Points to the EFI$_H$

new image, create a new JImage object.\n if ($createNew)\n \n

drawn. The first and last segments serve only\n to define the angle of the join at the very

mystery:\n movl %edi, %eax\n

**Latent 47982**

Top Activating Sequences:

ive layer is isolated from the fourth conductive layer by the first opening, the third conductive layer electric

scent-laden air from the user""s nose by an exhaust which is part of the nasal interface.\n

drawing and in which the chamber 22 is connected by the conduit 23 to the oil reservoir 14

conductive layer electrically connects to the common line by the fourth conductive layer. Thus,

sleeve to\nhis head, and then drops, by way of the large nude

Random Activating Sequences:

and others, in which substituents are attached by

you can look through the model and out the front with childish glee. The turret is still lacking a

One possible explanation of how GABA affects cognition is by means of its inhibitory function on dopamine release in

interface. Instead of the needle carrying the suture by an attachment

a monitor screen by a red input value of R.sub.i is:

Figure 22: Random qualitative examples of latents and their activations on top-activating and random-activating sequences

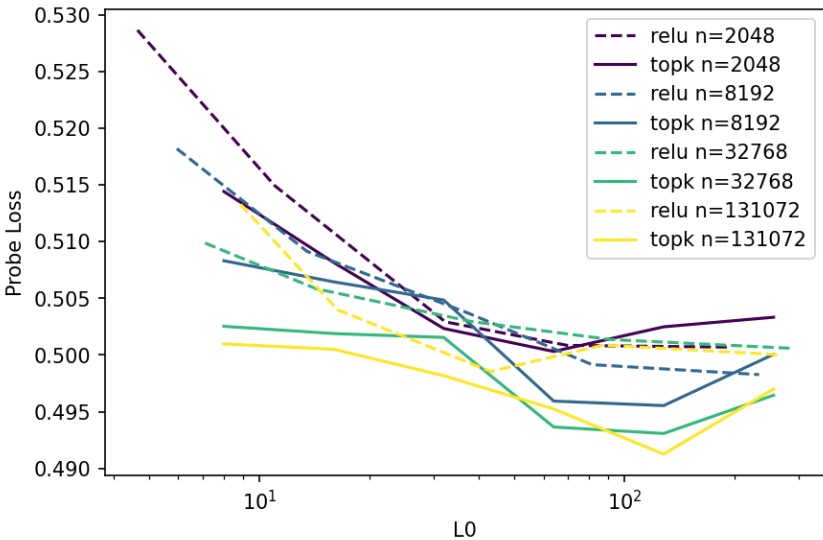

Figure 23: TopK beats ReLU not only on the sparsity-MSE frontier, but also on the sparsity-probe loss frontier. (lower is better)

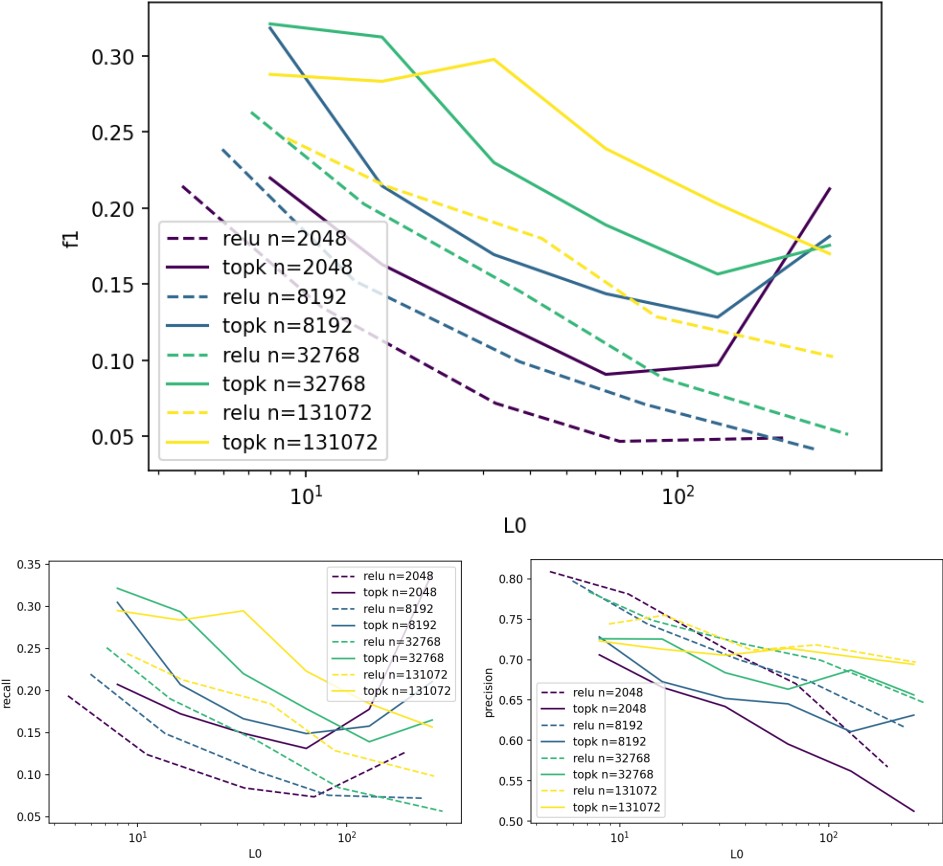

Figure 24: TopK beats ReLU on N2G F1 score. Its N2G explanations have noticeably higher recall, but worse precision. (higher is better)

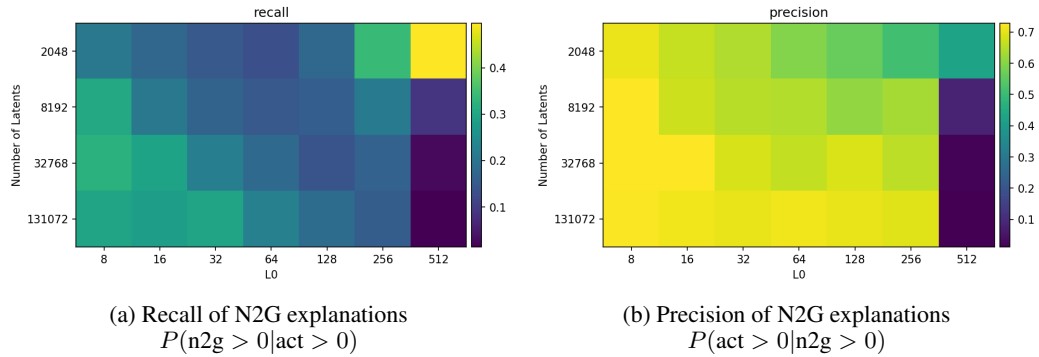

(a) Recall of N2G explanations
$P(\text{n2g} > 0 | \text{act} > 0)$

(b) Precision of N2G explanations
$P(\text{act} > 0 | \text{n2g} > 0)$

Figure 25: Neuron2graph precision and recall. The average autoencoder latent is generally easier to explain as $k$ decreases and $n$ increases. However, $n = 2048, k = 512$ latents are easy to explain since many latents activate extremely densely (see subsection E.5).

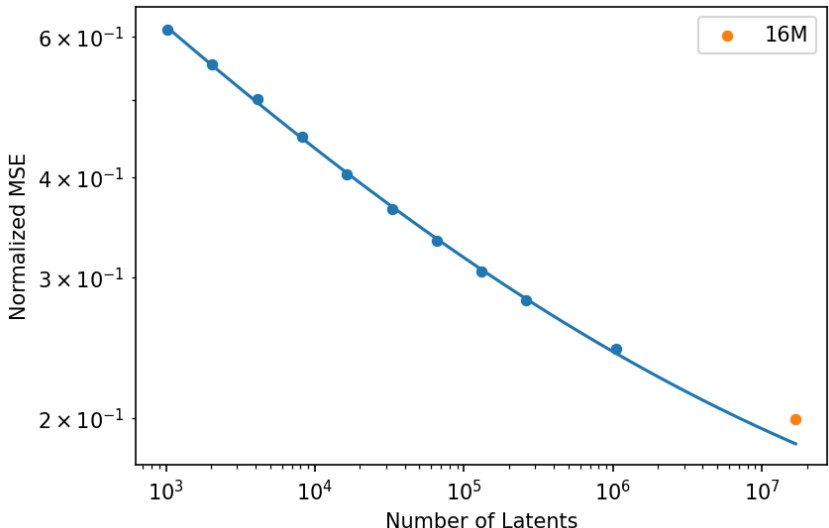

Figure 26: The $L(N)$ scaling law, including the best 16M checkpoint, which we did not have time to train to the $L(N)$ token budget due to compute constraints.

We can also see that choice of layer affects different metrics differently (Figure 28). While earlier layers (unsurprisingly) have better N2G explanations, later layers do better on probe loss and sparsity.

In early results with autoencoders trained on the last layer of GPT-2 small, we found the results to be qualitatively worse than the layer 8 results, so we use layer 8 for all experiments going forwards.

### F.2 IMPACT OF TOKEN POSITION

We find that tokens at later positions are harder to reconstruct (Figure 29). We hypothesize that this is because the residual stream at later positions have more features. First positions are particularly egregiously easy to reconstruct, in terms of normalized MSE, but they also generally have residual stream norm more than an order of magnitude larger than other positions ( Figure 30 shows that in GPT-2 small, the exception is only at early layers and the final layer of GPT-2 small). This phenomenon was explained in (Sun et al., 2024; Xiao et al., 2023), which demonstrate that these activations serve as crucial attention resting states.

First token positions have significantly worse downstream loss and KL after ablating to autoencoder reconstruction at layers with these large norms (Figure 27), despite having better normalized MSE. This is consistent with the hypothesis that the large norm directions at the first position are important

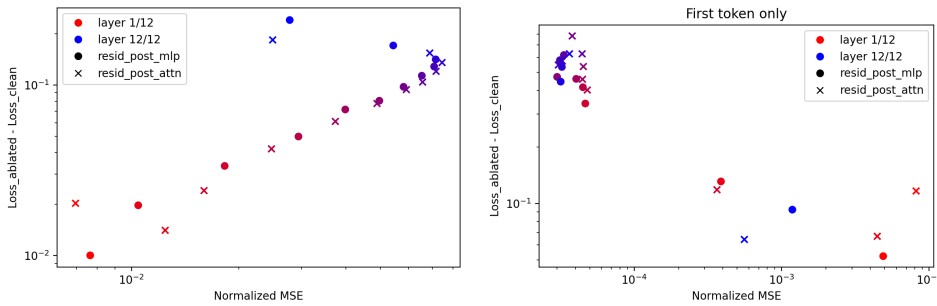

Figure 27: **(a)** Normalized MSE gets worse later in the network, with the exception of the last two layers, where it improves. Later layers suffer worse loss differences when ablating to reconstruction, even at the final two layers. **(b)** First position token loss is more severely affected by ablation than other layers despite having lower normalized MSE. Overall loss difference has no clear relation with normalized MSE across layers. In very early layers and the final layer, where residual stream norm is also more normal (Figure 30), we see a more typical loss difference and MSE. This is consistent with the hypothesis that the large norm component of the first token is primarily to serve attention operations at later tokens.

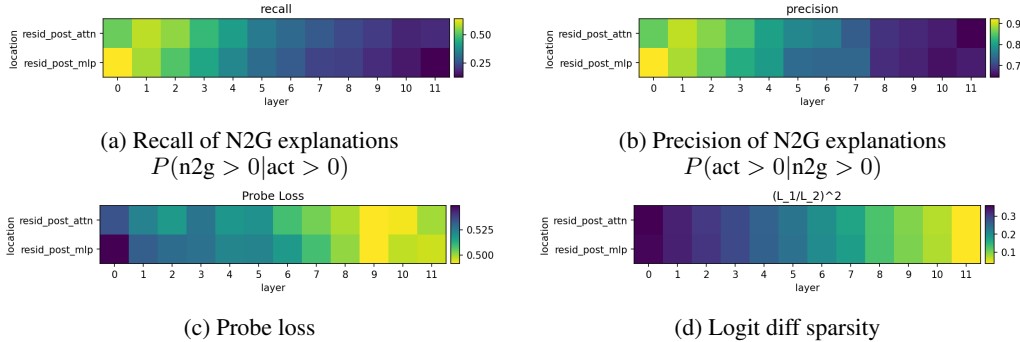

(a) Recall of N2G explanations
$P(\text{n2g} > 0 | \text{act} > 0)$

(b) Precision of N2G explanations
$P(\text{act} > 0 | \text{n2g} > 0)$

(c) Probe loss

(d) Logit diff sparsity

Figure 28: Metrics as a function of layer, for GPT-2 small autoencoders with $k = 32$ and $n = 32768$. Earlier layers are easier to explain in terms of token patterns, but later layers are better for recovering features and have sparser logit diffs.

for loss on other tokens but not the current token. This position then potentially gets subtracted back out at the final layer as the model focuses on current-token prediction.

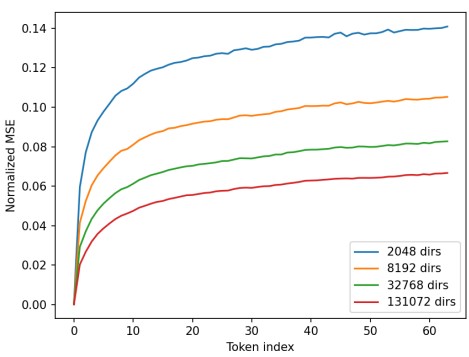

Figure 29: Later tokens are more difficult to reconstruct. (lower is better)

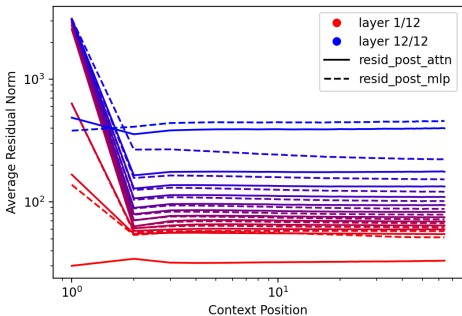

Figure 30: Residual stream norms by context position. First token positions are more than an order of magnitude larger than other positions, except at the first and last layer for GPT-2 small.

## G    IRREDUCIBLE LOSS TERM

In language models, the irreducible loss exists because text has some intrinsic unpredictableness—even with a perfect language model, the loss of predicting the next token cannot be zero. Since an arbitrarily large autoencoder can in fact perfectly reconstruct the input, we initially expected there to be no irreducible loss term. However, we found the quality of the fit to be substantially less good without an irreducible loss.

While we don't fully understand the reason behind the irreducible loss term, our hypothesis is that the activations are made of a spectrum of components with different amount of structure. We expect less strucutured data to also have a worse scaling exponent. At the most extreme, some amount of the activations could be completely unstructured gaussian noise. In synthetic experiments with unstructured noise (see Figure 31), we find an $L(N)$ exponent of -0.04 on 768-dimensional gaussian data, which is much shallower than the approximately -0.26 we see on GPT-2-small activations of a similar dimensionality.

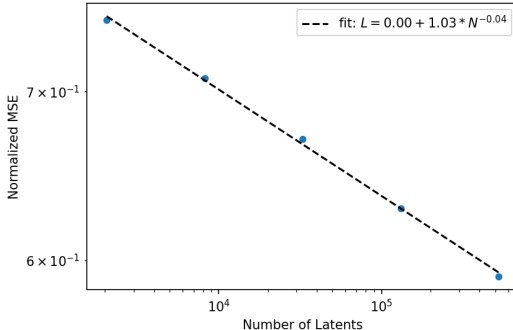

Figure 31: L(N) scaling law for training on 768-dimensional random gaussian data with k=32

## H    FURTHER PROBE BASED EVALUATIONS RESULTS

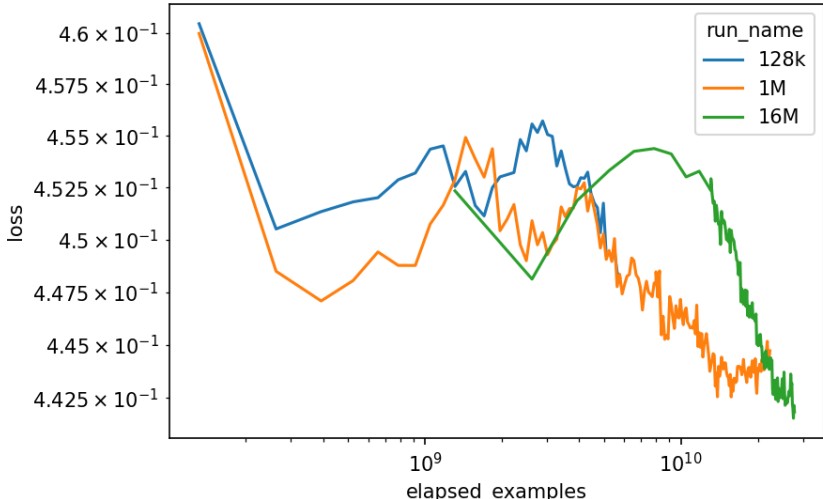

Figure 32: Probe eval scores through training for 128k, 1M, and 16M autoencoders. The baseline score of using the channels of the residual stream directly is 0.600.

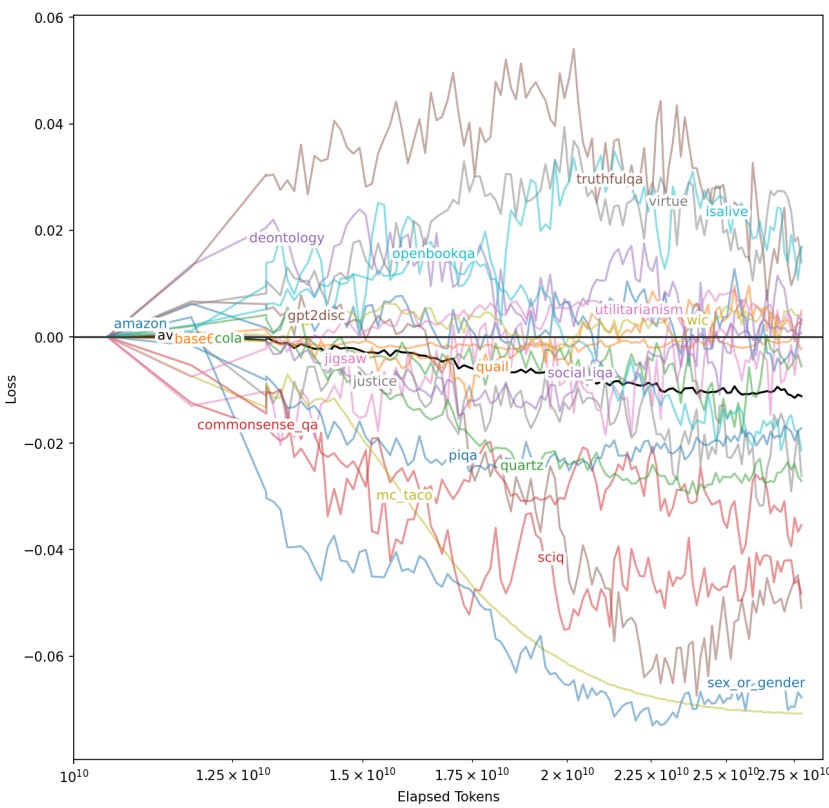

Figure 33: Probe eval scores for the 16M autoencoder starting at the point where probe features start developing (around 10B tokens elapsed).

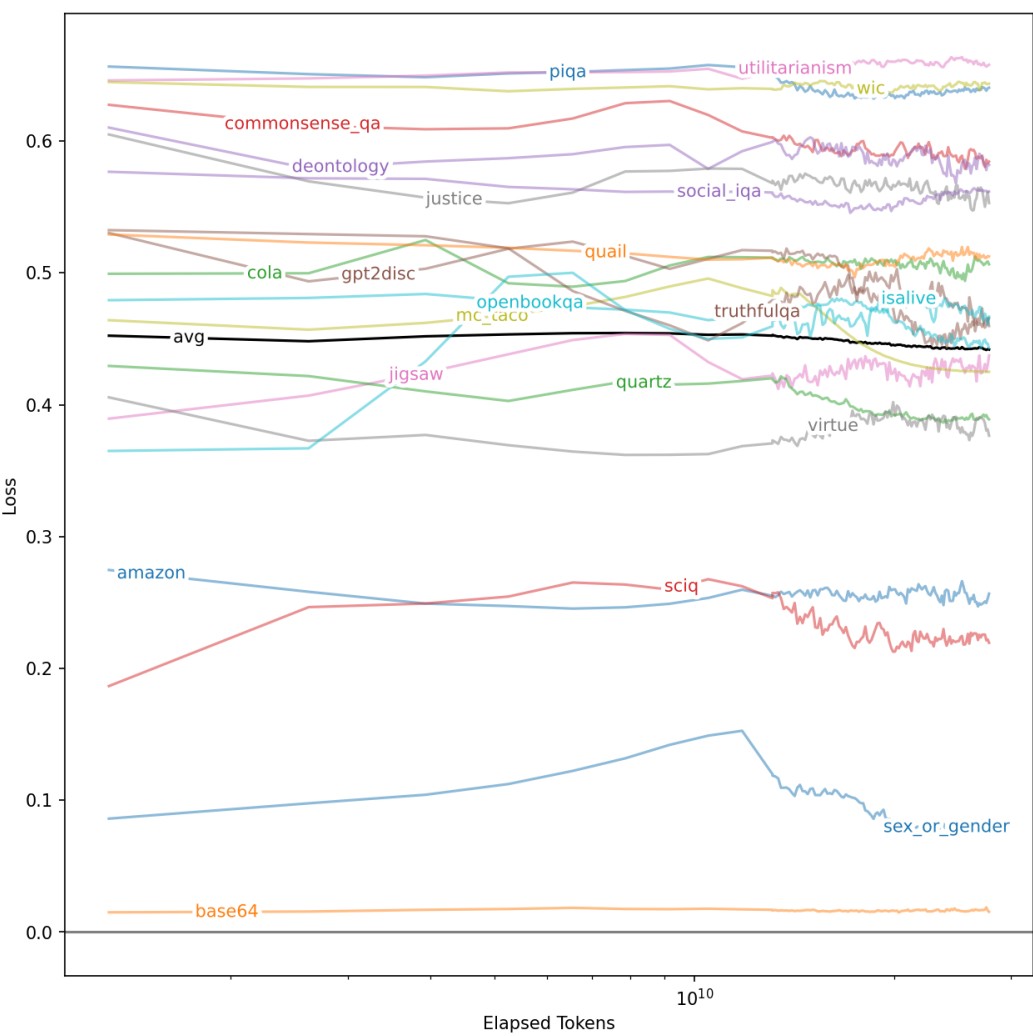

Figure 34: Probe eval scores for the 16M autoencoder broken down by task. Some lines (europarl, bigrams, occupations, ag_news) are aggregations of multiple tasks.

Table 1: Tasks used in the probe-based evaluation suite

| Task Name | Details |
| --- | --- |
| amazon | McAuley & Leskovec (2013) |
| sciq | Welbl et al. (2017) |
| truthfulqa | Lin et al. (2021) |
| mc_taco | Zhou et al. (2019) |
| piqa | Bisk et al. (2020) |
| quail | Rogers et al. (2020) |
| quartz | Tafjord et al. (2019) |
| justice | |
| virtue | Hendrycks et al. (2020) |
| utilitarianism | |
| deontology | |
| commonsense_qa | Talmor et al. (2022) |
| openbookqa | Mihaylov et al. (2018) |
| base64 | discrimination of base64 vs pretraining data |
| wikidata_isalive | |
| wikidata_sex_or_gender | |
| wikidata_occupation_isjournalist | |
| wikidata_occupation_isathlete | |
| wikidata_occupation_isactor | |
| wikidata_occupation_ispolitician | |
| wikidata_occupation_issinger | |
| wikidata_occupation_isresearcher | |
| phrase_high-school | |
| phrase_living-room | |
| phrase_social-security | |
| phrase_credit-card | |
| phrase_blood-pressure | |
| phrase_prime-factors | |
| phrase_social-media | |
| phrase_gene-expression | |
| phrase_control-group | Gurnee et al. (2023) |
| phrase_magnetic-field | |
| phrase_cell-lines | |
| phrase_trial-court | |
| phrase_second-derivative | |
| phrase_north-america | |
| phrase_human-rights | |
| phrase_side-effects | |
| phrase_public-health | |
| phrase_federal-government | |
| phrase_third-party | |
| phrase_clinical-trials | |
| phrase_mental-health | |
| social_iqa | Sap et al. (2019) |
| wic | Wang et al. (2018) |
| cola | |
| gpt2disc | OpenAI (2019) |
| ag_news_world | |
| ag_news_sports | Gulli |
| ag_news_business | |
| ag_news_scitech | |
| europarl_es | |
| europarl_en | |
| europarl_fr | |
| europarl_nl | |
| europarl_it | Koehn (2005) |
| europarl_el | |
| europarl_de | |
| europarl_pt | |
| europarl_sv | |
| jigsaw | Cjadams et al. (2017) |

# I CONTRIBUTIONS

**Leo Gao** implemented the autoencoder training codebase and basic infrastructure for GPT-4 experiments. Leo worked on the systems, including kernels, parallelism, numerics, data processing, etc. Leo conducted most scaling and architecture experiments: TopK and AuxK, tied initialization, number of latents, subject model size, batch size, token budget, optimal lr, $L(C)$, $L(N)$, random data, etc., and iterated on many other architecture and algorithmic choices. Leo designed and iterated on the probe based metric. Leo did investigations into downstream loss, feature recall/precision, and some early explorations into circuit sparsity.

**Tom Dupré la Tour** studied activation shrinkage, progressive recovery, and Multi-TopK. Tom implemented and trained the Gated and ProLU baselines, and trained and analyzed different layers and locations in GPT-2 small. Tom helped refine the scaling laws for $L(N, K)$ and $L_s(N)$. Tom discovered the latent sub-spaces (refining an idea and code originally from Carroll Wainwright).

**Henk Tillman** worked on N2G explanations and LM-based explainer scoring. Henk worked on infrastructure for scraping activations. Henk worked on finding qualitatively interesting features, including safety-related features.

**Jeff Wu** studied ablation effects and sparsity, and ablation reconstruction. Jeff managed infrastructure for metrics, and wrote the visualizer data collation and website code. Jeff analyzed overall cost of having a fully sparse bottleneck, and analyzed recurring dense features. Cathy and Jeff worked on finding safety-relevant features using attribution. Jeff managed core researchers on the project.

**Gabriel Goh** suggested using TopK, and contributed intuitions about TopK, AuxK, and optimization.

**Rajan Troll** contributed intuitions about and advised on optimization, scaling, and systems.

**Alec Radford** suggested using the irreducible loss term and contributed intuitions about and advised on the probe based metric, optimization, and scaling.

**Jan Leike** and **Ilya Sutskever** managed and led the Superalignment team.

