# OpenReview forum: "Scaling and evaluating sparse autoencoders"
_ICLR.cc/2025/Conference — ICLR 2025 Oral_

### Official Review · Reviewer_PAw6 · 2024-10-24

**Soundness:** 4
**Presentation:** 3
**Contribution:** 4
**Rating:** 10
**Confidence:** 4

**Summary:**

This paper (1) introduces Top-k sparse autoencoders (SAEs), which are novel for language models; (2) trains them on language models ranging in size up to GPT-4; (3) measures the SAE's sparsity/reconstruction tradeoff, finding performance that exceeds the state-of-the-art by a small margin; (4) demonstrate scaling laws for reconstruction accuracy in terms of training FLOPS and SAE width; and (5) measures the SAEs on a range of "downstream" metrics.

**Strengths:**

- This paper makes multiple novel contributions to research on the application of SAEs to language models:
	- The Top-k SAE architecture has not previously been used on language models. (Though they were introduced previously by (Makhzani & Frey, 2013).)
	- Scaling SAEs to a model the size of GPT-4 is a challenge due to the increasing concentration of dead features, and the early works in this field (Cunningham et al, 2023; Bricken et al, 2023) focused on relatively small language models. The scaling done in this paper is not entirely novel, as (Templeton et al, 2024) trained SAEs on the similarly-sized Claude 3 Sonnet, but this paper likely represents contemporaneous, parallel work.
	- Similarly, this paper finds scaling laws that are novel except for (Templeton et al, 2024).
	- This paper investigates several methods for evaluating SAEs beyond sparsity and reconstruction accuracy, which are important for establishing outside validity and utility. Especially appreciated is the metric in lines 311-314, contextualizing downstream loss in terms of a fraction of pre-training compute.

- In addition to the quantitative results, Top-k SAEs provide qualitative benefits over previous SAE architectures:
	- The sparsity level of Top-k SAEs is directly set, instead of relying on an indirect hyperparameter.
	- Top-k SAEs do not suffer from "feature shrinkage" caused by L1-regularization. The authors clearly demonstrate this in the excellent Figure 8.
	- Top-k SAEs have far fewer dead features at large scales.

**Weaknesses:**

- While the authors describe many results, they are not all equally justified.
	- Section 4.2 (Recovering Known Features with 1D Probes) is conceptually very exciting, but somewhat confusing in its details.
		- The magnitude of the probe loss improvement seems very low, from ~0.52 in the worst case to ~0.49 in the best case. Figures 32 and 33 indicate that a similar magnitude of improvement across various run sizes and datasets. This section would therefore benefit from more baselining: what is the best such classifier based on a residual stream neuron? On a linear  probe trained on all the residual stream neurons? On similarly-sized noise?
		- The only such baseline in the paper is that in Figure 32 the caption says that the loss on residual stream channels is 0.600. But the figure shows the loss improving from 0.46 to 0.44 over training, so it seems like almost all of the gain of the SAE occurred in the first <1% of training.
		- The probe losses appear to be averaged across the 61 datasets. This should be clearly stated in the body of the text.
	- Figure 7 is hard to understand, and would benefit from more explanations of its significance.

- There are several typos in the body of the text (see below)

**Questions:**

If it is permitted for authors to make revisions before the final submission, there are several small changes that could improve the quality of the paper:

- The paper would benefit from a thorough proofreading for typos. Examples include:
	- Line 129 reads: "It removes the need for the penalty. is an imperfect approximation of , and it introduces a bias of shrinking all positive activations toward zero (subsection 5.1)."
	- In line 327, the expression "min E [y log σ(wzi +b)+(1−y)log(1−σ(wzi +b))]" must be negated to get entropy.
	- Line 367-368 is missing a word after "similar": "with the same n (resulting in better F1 scores) and similar (Figure 24)."
	- Line 384 "trie" should be "tree".

- In section 2.4, it would be good to describe the provenance of the technique "we initialize the encoder to the transpose of the decoder". This is described in (Conerly, 2024) (https://transformer-circuits.pub/2024/april-update/index.html#:~:text=in%20most%20cases).-,%F0%9D%91%8A,.,-The%20rows%20of), which the authors should cite as inspiration or an independent replication. The other technique in section 2.4, auxiliary loss, is properly cited to (Jermyn & Templeton, 2024) in Appendix A.2.

- (Uncertain) The paper's introduction links to "code and autoencoders for open-source models" and a "visualizer". In the reviewer copy, these links are broken, likely to keep the authors anonymous. The authors should confirm that these links work in the final submission version.

---

> ### Author Response · Authors · 2024-11-24
>
> We thank the reviewer for the detailed review.
> - For the probe loss baselines: the residual stream neuron baseline performs much worse; we do not present results on probes trained on the entire residual stream, but they perform substantially better than single latents.
> - For N >> d, it’s intuitive that the best probe across N random directions will be better than the best probe across the d basis directions, especially since the residual stream doesn’t have a privileged basis (so the basis is arbitrary), so it would make sense for the loss to start out better than the single-channel baseline.
> - We will add a sentence to the camera ready clarifying that the scores reported are averaged across probing datasets.
> - We can add some additional interpretation of figure 7 to the main body.
> - There was a latex error that caused all instances of L0 and L1 in the main text to go missing; this will be fixed in the camera ready.
> - Line 384 "trie" is correct (https://en.wikipedia.org/wiki/Trie)
> - We will cite Conerly (2024) in section 2.4 as concurrent work.
> - The links are indeed broken for anonymization reasons and will be fixed in the camera ready version.

---

> > ### Comment · Reviewer_PAw6 · 2024-11-25
> >
> > Thank you for your response! If you are able to incorporate these changes, including a good explanation of figure 7, I would be able to increase my confidence to a 5.
> >
> > To summarize my thoughts on this paper, this work is one of the two most important and exciting papers on the subject of sparse autoencoders this year (the other being Templeton et al, 2024). It addresses many of the potential obstacles to applying an SAE to a production-scale language model which loomed large in the past 12 months:
> > 1. The paper demonstrates that scaling is achievable with relatively few dead latents.
> > 2. The paper demonstrates that reconstruction error varies smoothly with training (via scaling laws).
> > 3. The paper provides a range of much-needed evaluation metrics for the quality of the SAE, though all have a shortcoming such as being computationally expensive or relying on significant assumptions.
> > 4. Additionally, the paper innovates by applying the top-k architecture to SAEs, which performs at the frontier of reconstruction/sparsity, while having qualitative improvements like removing feature shrinkage and allowing the direct setting of the sparsity level.
> >
> > Although the paper has some flaws, as highlighted by other reviewers, I believe it deserves to be not only accepted, but highlighted at the conference.

---

> > > ### Author Response · Authors · 2024-11-26
> > >
> > > We thank the reviewer for the response. We have uploaded a new version of the paper that makes the requested changes, including an updated caption for figure 7:
> > >
> > > > Downstream loss on GPT-2 with various residual stream interventions at layer 8.  When every latent in the autoencoder is explained with N2G and the model is run using the N2G simulations instead of the real latent activations (purple), we find that larger and sparser autoencoders have better downstream loss. In the best case, running the model with N2G simulated latents performs better than a baseline of bigram language modeling, showing some nontrivial degree of explanation. However, most of the original autoencoder's reconstruction (red) remains unexplained by N2G explanations.

---

### Official Review · Reviewer_c1TT · 2024-10-29

**Soundness:** 3
**Presentation:** 4
**Contribution:** 4
**Rating:** 8
**Confidence:** 4

**Summary:**

In this paper, the authors scale and evaluate sparse autoencoders (SAEs) on modern large language models (LLMs) across different model sizes (up to GPT4).

The main contributions are as follows:
- SAEs are trained and evaluated across different model sizes, and scaling laws are found.
- The top-k activation function is leveraged to obtain state of the art results, and its ability to avoid activation shrinkage is explored
- An auxiliary loss is devised, and it significantly reduces the amount of dead features
- New metrics are introduced for evaluation: the explanation loss (obtained using just interpretable explanations), and the probe loss (where the authors check wether the SAE reproduces known features).

My **recommendation**: Accept, 8/10 (good paper).

### Why not lower score?

- Scaling SAEs was a very important problem in interpretability, as indicated in the seminal SAE paper.
- Top-k SAEs are a significant improvement on an important research topic. This approach is well motivated and well placed in the literature.

### Why not higher score?

- The performance of top-k SAEs is state of the art, but not beyond comparison with alternatives.
- While top-k SAEs are a significant incremental improvement, top-k SAEs are not new, and this is a new application.

**Strengths:**

The paper's strengths are as follows:

## Scale:

LLMs are being used in an increasing number of fields, and research into LLM interpretability is important for ensuring their fairness and safety, especially at scale.
This paper scales up SAEs to frontier LLMs, such as GPT4, which is a ***significant improvement*** over previous work, which was limited to smaller models. Furthermore, scaling laws are found, which could be very useful to practitioners and researchers trying to predict SAE performance ex-ante.

## Top-K:

The paper's usage of top-k SAE is one of the main contributions of this work, and it is ***original*** to the extent that top-k sparse autoencoders hadn't (to the best of my knowledge) been used on LLMs before. The ***quality*** of the results of the top-k function is such that it represent the state of the art when compared to other activation functions.

## Auxiliary Loss:

The authors tackle the ***significant*** issue of dead latents (up to 90% of the total when not mitigated) by introducing an auxiliary loss to the overall loss function. This incentivises the model to use all the latents, as it also supervises the next-aux latents. The ***quality*** of the results is high, as it reduces dead latents by an order of magnitude to 7%. This is to the best of my knowledge an ***original*** contribution.

## Evaluations:

The paper introduces many new evaluations. I believe that explainable loss and known latent probing are the most ***significant***.
Explainable loss is an ***original*** metric that considers the loss from just using the explanations that the SAE generates (discarding uninterpretable latents), and it represents a ***significant*** improvement over either MSE or L0 alone since it includes both reconstruction quality and an interpretability guarantee (L0 does not). Known latent probing is interesting in that it shows that the ***quality*** of probes of the latents is 25% better than that of residual stream probes.

## Clarity:

The paper is very ***clear*** across all sections, and its presentation is of high ***quality***

**Weaknesses:**

The paper's weaknesses are as follows:

## Irreducible error:
The paper notes that adding an irreducible error term to the scaling laws improves the laws. However, the explanation of why such an error term would exist in LLM activations is not clear, and an experiment showing that gaussian noise is hard to fit for SAEs is provided instead. The experiment does explain why such an error would be there in the first place, and its outcome is not surprising. Mentioning "components with different amounts of structure" is vague.

## Top-K vs penalty shrinkage experiments:
It is expected that having larger activations for top-k SAEs won't improve the loss and that it will for sparsity penalty SAEs. This is because top-k SAEs do not have any constraint on the magnitude of the k activations they use. An experiment on shrinkage does not make the top-k activation function more attractive.

## Reproducibility:

An issue with the experiments in this paper is that GPT4 is a closed source model, which makes it hard to reproduce the results of some of the experiments.

## Probing datasets:

A set of 61 binary classification tasks is used to train probes on the latents. It is not clear why these specific tasks have been chosen. A larger and more diverse set of tasks would have been more significant.

## Unclear phrasing of top-k's contributions:
In line 132 there is a missing word after: "instead of", which I assume to be L1, but it's unclear.

**Questions:**

How much better is the aux loss than the top-k loss?
This is interesting as it quantifies how much reconstruction quality we are 'paying' for sparsity.

At lines 132 and 452 do you believe 'an' should be 'a' (or is it not a typo)?

The following refers to work that is contemporaneous (4 months old or less), and it does not affect my rating.
Are you aware of:
- Jumprelu SAEs?
- Mixture of Experts SAEs? I believe you hinted at this in the paper.
- Sparse feature circuits? And the possibility to do evals on known circuits not just known latents (SAE seem not to be the best at factual recall).

---

> ### Author Response · Authors · 2024-11-24
>
> We thank the reviewer for the detailed review.
> - The intuition for the irreducible loss term is that SAEs can only efficiently model the distribution if there is some structure (i.e there are features that compose together to explain the distribution well). If there is no structure, then it is very inefficient to model the distribution by increasing the number of latents. When we say “components with different amounts of structure”, what we mean is that the residual stream might be a linear combination of components that scale differently with number of latents due to different amounts of composability.
> - We agree that the lack of shrinkage in topk is not surprising. Our goal in section 5.1 is just to double check this prediction, and to see whether this explains the entire difference between topk and relu.
> - The probing datasets were chosen somewhat arbitrarily, based on a subjective sense of what kinds of features were likely to occur. We’re excited about future work further expanding the size and diversity of the set of tasks.
> - There was a latex error that caused all instances of L0 and L1 in the main text to go missing; this will be fixed in the camera ready.
> - The aux loss is often hard to interpret because as the main loss improves, the task represented by the aux loss changes. To see how much reconstruction quality we are paying for sparsity, we can compare autoencoders with different k (see figure 1b).
> - We’re excited about the concurrent and subsequent work on jumprelu, MoE SAE, and sparse feature circuits. We can include citations to work on these directions in the related work section for the camera ready.

---

> > ### Comment · Reviewer_c1TT · 2024-11-25
> >
> > Dear authors,
> > Many thanks for your detailed response.
> > Regarding your points:
> > - I understand what the irreducible error means, but my point is that it’s unclear why there would be random noise in the model’s activations. Is it that the model is underused? Unlikely since it’s in superposition. Are the structures in the activations not linear?
> > - I trust you will fix the latex error.
> > - I agree that varying k is also a good measure of how much reconstruction we pay for sparsity. This is effectively like checking the next k latents.

---

> > > ### Author Response · Authors · 2024-11-26
> > >
> > > One intuition for why there might be noise is that the minibatch randomness and initialization introduce noise into the weights, so that different inputs which should activate the same features might have slightly different activations (along dimensions which have low impact on the final loss) due to using different circuits. There is also definitely the possibility of other kinds of nonlinear structures.
> > >
> > > Also, even if all the structures are linear, the degree of compositionality will affect the slope of the scaling law too. For example, the model might represent features for dog breeds with size features composed with length of hair features composed with hair. color features etc, so that dogs with similar attributes will cluster together; or it might have a golden retriever feature and a poodle feature and so on completely arbitrarily distributed. In the latter case, the improvement in reconstruction from adding additional features is much smaller.

---

### Official Review · Reviewer_Zr1S · 2024-11-04

**Soundness:** 4
**Presentation:** 3
**Contribution:** 4
**Rating:** 10
**Confidence:** 4

**Summary:**

The paper develops training methodology and scaling laws for a new sparse autoencoder (SAE) variant - TopK SAEs, which replaces the L1 penalty in the standard SAE (https://transformer-circuits.pub/2023/monosemantic-features) with a TopK filter, so that the desired sparsity of activations can be set directly. The new architecture is evaluated against other popular SAE variants along the reconstruction-sparsity frontier. The interpretability of the TopK SAE latents is also measured via several novel metrics.

In more detail:
- the TopK filter overcomes the shrinkage problem associated with the L1 penalty, allowing better reconstruction with the same sparsity;
- to minimize dead latents, the TopK training method uses a simple initialization strategy and an auxiliary loss that incentivizes the dead latents to reconstruct the error of the SAE.
- scaling laws for TopK are studied w.r.t. various parameters, such as the number of latents, the $k$ value, and the size of the model whose activations SAEs are trained on.
- several evaluations beyond MSE/sparsity tradeoffs are performed:
	- downstream loss when SAE reconstructions are patched in
	- using individual SAE latents as probes for binary classification tasks
	- simple autointerpretability based on Neuron to Graph (N2G)
	- evaluating how sparse the effect of ablating SAE latents is on logits of the model

**Strengths:**

- the paper presents a new, simpler SAE architecture and training stack that demonstrably scales to frontier models and many SAE latents. The main advantages over concurrent work are:
	- the sparsity can be set directly, making it easier to do hyperparameter tuning;
	- the training recipe is also simple and with few knobs to tune;
	- yet results on the MSE/sparsity frontier are state of the art, and there are very few dead latents
- many of the findings are quite well supported by extensive comparisons with other SAE variants
- the discussion on the limitations of autointerpretability methods in 4.3. is illuminating
- the paper proposes some new and interesting interpretability evaluations, especially probing and sparsity of downstream effects

**Weaknesses:**

- as the authors are already aware, using individual SAE latents as probes for very high-level binary tasks (such as sentiment) may be too restrictive, especially in light of phenomena such as feature splitting. This limits the significance of the results
- another practical limitation of the TopK architecture is that it forces every activation to use exactly $k$ active features. In practice (at least in my experience) this may lead to many unimportant features to activate. See concurrent work https://www.alignmentforum.org/posts/Nkx6yWZNbAsfvic98/batchtopk-a-simple-improvement-for-topk-saes
- the interpretability evaluation is at times too reliant on high-level aggregate metrics. For instance, how should we interpret the "sparsity of downstream effects" numbers?
- the presentation on scaling laws would benefit from a comparison with concurrent work by Anthropic: https://transformer-circuits.pub/2024/scaling-monosemanticity/#scaling-scaling-laws
- there is some inconsistent notation (e.g., using $N$ and $n$ for the number of latents) and some missing symbols from the text

**Questions:**

- did you use the scaling laws to choose hyperparameters for the largest SAE training runs? Do you know how well that worked?
- Have you thought about ways to more scalably evaluate precision of SAE latent explanations?

---

> ### Author Response · Authors · 2024-11-24
>
> We thank the reviewer for the detailed review.
> - We agree that feature splitting limits the usefulness of probes on individual latents; as a result, we chose to consider latents before applying the activation function, so that split features would still correlate strongly with our labels. In some unpublished experiments we explored the qualitative interpretation of the chosen latents but found that there was a tradeoff between higher qualitative quality of latents when considering post-activation, and greater robustness to feature splitting. We omitted this discussion from the paper because our results were relatively preliminary.
> - We agree that restricting to exactly k active features is not ideal, and we’re excited about follow-up/concurrent work like batch topk and jumprelu. We also have some results on replacing the topk activation function at test time with a jumprelu with only a moderate hit to the MSE-L0 frontier. We can include these results in the camera ready. We can use N instead of n everywhere for number of latents.
> - While we focus mostly on aggregated quantitative metrics, because we are interested in choosing between different autoencoder training setups, we are also excited about future work on more detailed qualitative investigation of these metrics. We do present qualitative results for some metrics, such as the n2g explainability metric.
> - There was a latex error that caused all instances of L0 and L1 in the main text to go missing; this will be fixed in the camera ready.
> - Yes, we used the scaling laws to choose hyperparameters for the largest runs. While it’s hard to know for sure whether the choice was optimal, our largest run appears to be on trend in our L(C) scaling law.
> - There are several directions for evaluating precision that we’re excited about - one approach is to use a LM to generate sequences matching the explanation, and adjusting for the distribution shift; another approach is to use cheap heuristics as a first stage filter before applying more expensive explanation simulation techniques.

---

> > ### Comment · Reviewer_Zr1S · 2024-11-26
> >
> > Thanks for your detailed responses. I believe you have addressed ~all my concerns in one form or another. Regarding comparisons to related works, I understand that this was likely concurrent work so I understand why this is not reflected in the paper.
> >
> > This paper has advanced the state of the art for training and evaluating SAEs, and has my strong recommendation for acceptance to ICLR 2025.

---

### Official Review · Reviewer_JgtD · 2024-11-04

**Soundness:** 4
**Presentation:** 4
**Contribution:** 4
**Rating:** 10
**Confidence:** 4

**Summary:**

The authors present a well written paper which makes multiple significant contributions to the field of language interpretability. The paper does not explicitly pose a set of questions, but the subject matter implies several concerns:
1. Do sparse autoencoders scale to larger models? Does this require more latents than for larger models?
2. Can sparse autoencoder training be improved? For example, can it be made simpler via use of a top-K activation function?
3. How should we evaluate sparse autoencoders? Are there alternatives to the sparsity-reconstruction pareto-frontier framing?

The paper makes a number of contributions.
1. First, they propose a novel training recipe based on the use of the Top-K activation function for directly imposing sparsity in the sparse bottleneck layer. They evaluate their methods against similar proposed methods via the sparsity-reconstruction pareto-frontier framing and find their recipe is competitive with the best methods in the existing literature.
2. Second, they demonstrate scaling laws for their sparse autoencoder training recipe using the GPT4 family of models.
3. Finally, they introduce a number of additional metrics for evaluating sparse autoencoders including loss-based metrics, metrics which rely on labeled datasets, automated interpretability and ablation sparsity.

The strongest contribution of the paper is clearly the application of SAEs to exceptionally larger models, showing that sparse autoencoders scale to such models, but the paper also makes numerous other contributions around training and evaluation methods.

**Strengths:**

## Strengths by Section:

**Training Recipe**: The paper is unusually detailed with respect to the training recipe, which provides more clarity than similar publications about the training of sparse autoencoders. As noted by the authors, use of the Top-K activation function in a sparse autoencoder is not new, but it is novel in the context of sparse autoencoders trained on model activations. Importantly, it solves a previously documented issue called "shrinkage" which results in latent activations being biased towards activating less strongly than they theoretically should.

**Scaling laws**: The paper demonstrates not only the training of sparse autoencoders on a SOTA language model, substantiating claims that this technique is scalable. Use of GPT4 provides one of the strongest possible arguments that sparse autoencoders can scale to very large LLMs. Additionally, the authors provide clear scaling laws in terms of the compute, data and number of latents. Though possibly a minor detail, we appreciate that the authors highlighted their initial incorrect prediction that there would need to be non-zero irreducible loss (as such comments help inform readers that the result was not obvious and thus increases readability and makes the significance clearer).

**Evaluating SAEs**: Though plausibly it would be reasonable to separate work scaling sparse autoencoders from work exploring how best to evaluate them, we appreciate the attempt to rectify the current gap in the literature around how to evaluate sparse autoencoders. Contributions in this section are well motivated and varied. Section 4.3 is particularly valuable, as Neuron 2 Graph is a plausibly much cheaper way to perform automatic interpretability as opposed to the use of language models.

## Strength by Dimension

**Originality**: While the work mostly combines existing ideas from the literature placing them in the context of sparse autoencoders trained on model activations, this hardly diminishes the contribution. Some ideas may be more clearly novel than others (such as the pre-training compute based metric that enables better interpretation of the increase in downstream loss in section 4.1).

**Quality**: The work is generally high quality. There are no glaring issues that I can see.

**Clarity**: Details of both the methods and reasoning are generally high quality.

**Significance**: The paper is highly significant due to contributions along multiple lines (training recipe, scaling laws and evaluation of SAEs).

**Weaknesses:**

**Training Recipe:**
- We appreciate that the proposed method eliminates the need for finetuning of the L1 coefficient (though choice of the number of activated latents is still required). We might interpret later analysis as attempting to identify a more principled way for setting the number of activations (setting K), though in the absence of better motivation, directly setting the number of activating latents isn’t clearly valuable.
- Comparison of sparse autoencoder architectures is somewhat shallow, looking only at the sparsity-reconstruction pareto frontier. For example, the authors do not attempt to compare latents found by each method to establish whether they provide similar or distinct decompositions (is there often a 1:1 map between latents in sparse autoencoders trained via each method?).
- The authors could have provided demonstrations that top-K sparse autoencoders work on toy models of superposition (as has been previously done to establish reason to prefer one architecture over another).
- Whilst the authors consider alternative methods for evaluating sparse autoencoders later in the work, they should have revisited the training recipe comparison in the context of their other evaluation methods. It is unclear whether Gated / topK SAEs would be comparable on all evaluation methodologies.
- The authors miss one possible issue with top-K activation functions. Use of the Top-K activation function means that the activation of a given latent cannot be calculated independently of others which is a minor but real obstacle to working with very large autoencoders (for example, when identifying max activating examples, one must first calculate all latent activations, then select the top-K latents).
- Training on 64 token context sizes might affect the results. The models studied certainly can handle much longer prompts. The authors don’t note whether the SAEs generalize to activations produced during the process of larger numbers of consecutive tokens.

**Scaling Laws**. We have no specific comments on the weaknesses of this section.

**Evaluating SAEs**
- Though contributions in this section are valuable, there is an absence of discussion resolving possible metrics and identifying which are best to use in practice. Though many related papers likely deal with the current uncertainties of the field similarly (by presenting various possible motivations / metrics without meaningfully resolving which is best). We believe that devoting more of the discussion to comparing metrics and resolving insights found via each may be fruitful.
- (Repeating a point from earlier) The authors don’t evaluate non-TopK autoencoders via any of the new metrics leaving possibly valuable insights on the table.
- Section 4.2 could have used a linear probe on the model activations as a baseline against which to compare the 1d logistic probe results. The authors also don’t perform any error analysis or raise any particular examples of latents which clearly track expected features of the input. While such qualitative analysis may not scale, it might possibly provide much better intuition about the results. The authors don’t address how sparse combinations of latents might jointly represent features.
- The authors could attempt to compare Neuron 2 Graph explanations to those produced by language models (which is the norm for automatic explanations of features in the rest of the sparse autoencoder literature).

**Questions:**

Some work that might improve this paper:
- Checking whether top-K sparse autoencoders find the same latents as other methods. Reliance on summary statistics when making comparisons between architectures could be misleading.
- Establishing that top-K sparse autoencoders work well on toy models of superposition (and using this to better explain why top-K might be better than other architectures).
- Using longer prompts than 64 tokens when training sparse autoencoders (this seems rather short) or instead providing an empirical analysis showing that this hyperparameter has no meaningful impact on results.

Since my overall impression of the paper is quite good, and most of the weaknesses, if addressed, would substantially increase the length of the paper, I don't believe addressing any given issue is essential.

---

> ### Author Response · Authors · 2024-11-24
>
> We thank the reviewer for the detailed review.
> - One major advantage of being able to set the sparsity directly is it makes it much easier to do scaling laws where sparsity is held constant.
> - We’re excited for future work on whether top-k finds the same kinds of latents as other autoencoder methods, how top-k behaves on toy models of superposition, whether longer prompts than 64 tokens have better results.
> - We do briefly compare TopK and ReLU on probe loss and explainability in figures 23 and 24, but we’re excited about more in-depth future work on how different autoencoder methods perform on metrics other than reconstruction/sparsity.
> - To avoid the issue of TopK preventing values from being calculated independently, the TopK activation function can be replaced by a JumpReLU at test time chosen to target the same L0. This leads to a small increase in MSE (at comparable L0), but this increase can be largely reduced by training with multiple k values. These results were cut from the submission, but we can add them back in the camera ready.
> - We’re excited for future work that focuses on comparing and refining autoencoder metrics.
> - We generally find that linear probes on the entire residual stream are substantially better than the best probes on single autoencoder latents.
> - In some unpublished experiments we explored the qualitative interpretation of the chosen latents but found that there was a tradeoff between higher qualitative quality of latents when considering post-activation, and greater robustness to feature splitting. We omitted this discussion from the paper because our results were relatively preliminary.
> - We focus on k=1 because of its conceptual and computational simplicity, but we’re excited about future work on k-sparse probes for k > 1 for autoencoder evaluation.
> - While we don’t directly compare the explanations produced by n2g and auto interpretability in our paper, we do explore the auto explanation scores in figure 19. We generally find that auto interpretability scores are very noisy.

---

> > ### Comment · Reviewer_JgtD · 2024-11-28
> > **We thank the authors for replying to our comments / questions and will leave our rating as is.**
> >
> > We thank the authors for replying to our comments / questions and will leave our rating as is.

---

### Official Review · Reviewer_CUAP · 2024-11-04

**Soundness:** 3
**Presentation:** 3
**Contribution:** 2
**Rating:** 3
**Confidence:** 4

**Summary:**

The paper proposes the use of a k-sparse auto encoder model to  control sparsity, simplifying tuning and improving the reconstruction-sparsity frontier. The paper presents results showing better reconstruction and minimal dead latents. The authors trained a 16-million latent autoencoder on GPT-4 activations over 40 billion tokens, demonstrating clear scaling laws for the autoencoder size and sparsity. The main contribution of the paper is the use of TopK activation to limit the number of active latents and presenting metrics of latent quality.

This limiting factor seems to play a role in allowing scaling the model for larger encoders as demonstrated  experimentally on GPT2 with multiple ablation studies and an in-depth evaluation downstream loss, explainability amongst other metrics.

**Strengths:**

1. The adoption of k-sparse autoencoders with TopK activation is a significant improvement over traditional ReLU-based models. By directly controlling the number of active latents, TopK provides a straightforward way to balance reconstruction quality and sparsity, simplifying tuning and improving interpretability.

2. The study of scaling laws for sparse autoencoders across various dimensions—autoencoder size, language model size, and sparsity—is well-executed. This contributes valuable insights into how autoencoders behave at larger scales, especially when trained on large models like GPT-4. Clear scaling patterns provide guidelines for researchers to optimize model size and token budgets effectively.

3. The authors address the issue of "dead latents" effectively by using specific initialization and auxiliary loss techniques. This innovation minimizes computational waste and improves model efficiency, making it feasible to train much larger autoencoders while keeping most latents active, even at high scales.

4.The paper introduces thoughtful metrics that go beyond simple reconstruction errors to evaluate feature quality. Metrics like downstream loss, probe loss, explainability, and sparsity of downstream effects offer a more nuanced assessment of latent quality, highlighting the interpretability of features that the autoencoders extract. This focus on interpretability is particularly important for mechanistic understanding and applications in AI alignment.

5. The use of TopK avoids the activation shrinkage problem that commonly affects ReLU-based sparse autoencoders. This makes TopK particularly suitable for training sparse models without the need for additional regularisation to combat shrinkage, which simplifies model training and improves performance.

**Weaknesses:**

1. As highlighted in the limitations section, The TopK activation function might be overly restrictive. Each token must use exactly k latents, which could be suboptimal for capturing certain features that require more flexible sparsity levels based on input complexity.

2. The experiments primarily use a context length of 64 tokens, which may not capture longer dependencies present in language models like GPT-4. By not exploring larger context lengths, the paper might miss behaviors and interactions that occur over more extended sequences, which could be valuable for understanding latent patterns and achieving higher interpretability.

3. Although the paper introduces meaningful metrics for interpretability, some, like the downstream loss and explainability evaluations, require substantial computational resources, particularly for large models like GPT-4.

4. The reliance on an auxiliary loss to prevent dead latents may add extra complexity to the training process. While effective, this approach could complicate optimization and might be sensitive to tuning, making it more challenging to reproduce results consistently across different datasets and architectures.

5. Although the approach seem to scale on relatively smaller model, it is not clear how scalable this approach for much larger model.s

6. While the paper discusses applications like anomaly detection and mechanistic interpretability, there is limited empirical testing in these areas. Applying these sparse autoencoders to real-world tasks and evaluating their practical benefits would strengthen the claim of their utility and robustness across applications.

7. The explainability metric, which relies on methods like Neuron to Graph (N2G), is noted to have high recall but lower precision. This can create an "illusion of interpretability," where explanations seem broad but lack specificity. The paper briefly acknowledges this, but a more in-depth exploration of how to balance recall and precision would improve confidence in these metrics.

**Questions:**

1. Could you elaborate on the choice of fixed-k sparsity? Do you foresee any scenarios where adaptive sparsity (varying k based on input complexity) might improve model performance or interpretability? Have you experimented with such approaches?

2. Given that TopK activation controls sparsity effectively, do you see potential limitations of this method in real-world applications, where sparsity needs may vary dynamically?

3. Could you share more about how the interpretability metrics were validated? For instance, have you found that these metrics correlate well with human judgment in terms of interpretability?

4. How sensitive is the auxiliary loss to hyperparameter changes, particularly at larger scales? Could you share any guidelines or best practices for tuning this component?

---

> ### Author Response · Authors · 2024-11-24
>
> We thank the reviewer for the detailed review.
> - We think variable k sparsity is an exciting direction for future work. We did some initial explorations of this direction but ultimately did not include any in the paper.
> - We agree that experiments with a longer context length would capture more interesting features. We’re excited about future work exploring this in more detail.
> - It is true that some of the metrics presented require substantial computational power. We believe that it is nonetheless valuable to present metrics which are expensive. In addition, we also present inexpensive metrics (probe loss, logit diff sparsity) for use when using expensive metrics is infeasible.
> - The auxiliary loss represents a substantial simplification of methods used in previous work for dead latent prevention. Additionally, we find in practice that the optimization is not very sensitive to the auxiliary loss hyperparameters. We swept the auxiliary loss coefficient hyperparameter at small scale and found that it is not very sensitive. We believe that it is also not sensitive at large scale, but it would be prohibitively expensive to check. In appendix B.1 we give the recommendation to use a value of 1/32 for this coefficient.
> - As for scalability, we demonstrate that our approach scales smoothly to the largest existing language models (GPT-4).
> - We’re excited about future work demonstrating applications of sparse autoencoders to various tasks; however, it would be out of scope for this paper.
> - The illusion of interpretability effect occurs when the explainability is determined only on the basis of documents that activate the latent (i.e it only measures recall). We use N2G precisely because we want an explainability metric that measures both precision and recall.
> - Human judgment of interpretability is highly nontrivial and requires human interaction with the model in order to evaluate both precision and recall. We’re excited about future work exploring this direction but we felt it would be out of scope for our paper.

---

> ### Author Response · Authors · 2024-11-27
>
> Given that the deadline for making changes is fast approaching, please let us know if there are any changes we can make to address your concerns.

---

> > ### Comment · Reviewer_CUAP · 2024-11-28
> >
> > I address below the response in order:
> > 1. I believe this is such a crucial component that it needs to be in the paper. Otherwise the paper is a very good proof of concept but it is not mature enough yet for a publication. The proposed TopK is already seen as ineffective evident by this recent paper (https://openreview.net/pdf?id=d4dpOCqybL). Thanks to the public contributor for drawing my attention to their work.
> >
> > 2. The 64 tokens is seriously too low in the specific context of your paper around scalability so these results will not scale to the more recent models with thousands of tokens in length, therefore underminds the claimed generalisability of the scaling laws.
> >
> > 3. That is a reasonable argument, but this computational complexity is related to the point above as well. You would expect more work is needed to scale the metrics for larger context, again this questions the scalability claims of the paper.
> >
> > 4. Can you please provide a quantitative measure to “not very sensitive”? Running the sweeps at ‘a small scale’ weakens your recommendations. I advise to have the sweeps on smaller models with larger hyperparameter scale for the recommendations to be more actionable.
> >
> > 5. Please see my comments above.
> > .
> > 6. I believe you need to consider taking your approaches from a proof-of-concept to applying it on more testing scenarios so you have a better sense of the complexity of the problem.
> >
> > 7. That is a fair comment.
> >
> > 8. This is understood and I look forward to seeing your future work.

---

> > > ### Comment · Reviewer_c1TT · 2024-11-28
> > >
> > > Dear fellow reviewer.
> > >
> > > On your point 1.
> > > That work is contemporaneous under ICLR’s guidelines, and build on top of the work we are reviewing here. A further instance of work in this line is JumpReLU SAEs, as they take the work further by learning k.
> > >
> > > Since this paper came out sooner than JumpReLU and batch top-k, and both works were inspired by it, we should consider wether we want the impact this paper had after publication to be a positive factor or a negative factor in our evaluation.
> > >
> > > Sincerely, your fellow reviewer.

---

> > > > ### Comment · Reviewer_CUAP · 2024-11-28
> > > >
> > > > The other work was referenced as an additional data point that the topK approach is very strict and the variable k sparsity is a potential solution.
> > > >
> > > > I do not see how this paper came out sooner than batch top-k although batch top-k is already published while we are still reviewing this paper here? Considering this is a double-blind review process, I trust that this paper and any others addressing similar topics were developed independently by separate teams, without any intent to influence my impartial evaluation of this work. If these teams happen to be reviewing each other's papers, I trust that they will adhere to the highest standards of fairness and objectivity in their evaluations.

---

> > > > > ### Comment · Reviewer_c1TT · 2024-11-28
> > > > >
> > > > > Dear fellow reviewer.
> > > > >
> > > > > I said both came out later since they both cite this paper. I said they build on top of this paper since both make use of some variation top-k architecture, which I concede is not new in general, but this paper introduces in the context of interpretability in language modelling.
> > > > >
> > > > > I am in no way related to the team developing this paper, nor to any adjacent organisation. I do not know any of the authors, nor do they know me.
> > > > > Furthermore, I am in no way related to the teams behind the JumpReLU and Batch-topk sparse autoencoders.
> > > > >
> > > > > I can’t divulge any more details regarding my identity to preserve double blind, but I can assure you there is no conflict of interest in my comment.
> > > > > I am merely pointing out the ICLR guidelines (https://iclr.cc/Conferences/2025/ReviewerGuide), which specify that we should disregard contemporaneous work, and both JumpReLU and batch-topk are contemporaneous according to ICLR’s guidelines.
> > > > > If you feel like that is not fair or that those papers are not a significant factor in your evaluation feel free to disregard my comment.
> > > > >
> > > > > Sincerely, your fellow reviewer.

---

> ### Author Response · Authors · 2024-12-01
>
> Thank you for the follow-up comment.
>
> - We have some results that demonstrate that it is possible to replace the TopK at test time with a fixed threshold (thereby allowing variable sparsity); furthermore, it is possible to train on multiple k's at the same time to create autoencoders where k can be varied at test time. We are unfortunately past the deadline for updating the paper in this phase of reviewing, but if the ACs will allow it, we can include these results in the appendix for the camera ready. Our work was in fact conducted before the batch-topk work, and posted to arxiv and was not reviewed at NeurIPS. The batch-topk work was inspired by our work and introduces an incremental improvement; it does not demonstrate the ineffectiveness of topk, but rather that topk performs almost as well as batch-topk.
> - While it is certainly the case that longer contexts will involve more features, and it would have been ideal for us to also double check longer contexts in our paper, we believe that our experiments nonetheless provide strong evidence for scalability of our techniques. First, GPT-4 already exhibits highly nontrivial behavior within 64 tokens. Second, to explain the behavior on this context length required us to push the frontiers of sparse autoencoder training to train the largest sparse autoencoder (by # of alive latents) ever created, requiring the development of new scalable approaches. Third, it would be very surprising if there were very clean scaling laws at short context length, but no such clean scaling law at longer context lengths. Finally, it's not like GPT-4 is an ancient outdated model from an era when language models only supported 64 tokens; GPT-4 is a very recent model and models supporting thousands of tokens of context have existed for years, and we simply chose to only use 64 tokens of that context for reasons of convenience.
> - The computational cost of these metrics comes primarily from factors other than context length - for downstream loss, contexts on the order of thousands of tokens costs negligibly more (because you get the loss on every token in the context anyways), and for explainability, the main source of cost is rarity of the features, not context length.
> - Across an order of magnitude of choices of coefficient and auxk hyperparameters (4x4 runs in total), in one setting we explored, we found that the best and worst configurations had reconstruction losses varying between 0.0127 and 0.0122 (4% relative difference).

---

### Public Comment · ~Neel_Nanda1 · 2024-11-14
**Paper Thoughts**

Commenting as an unrelated mechanistic interpretability researcher, I consider this to be **one of the most important mechanistic interpretability papers of the past year, and worthy of being highlighted at the conference**. Sparse autoencoders seemed a highly promising technique, but had so far mostly been demonstrated on small-ish models, with limited proxy metrics, and had several known issues like shrinkage. This work:
* Efficiently scaled SAEs to GPT-4 - an extremely impressive engineering feat, and the furthest they have yet been scaled, along with contributing to our knowledge for how to efficiently train large SAEs with things like scaling laws. A key risk in mech interp is that the techniques used only work for small models, this work showed that was false for SAEs.
* Showed that the TopK architecture was an improvement, both in terms of performance and in terms of resolving issues like shrinkage, along with advantages like being able to set the L0. This is a practically useful finding for all SAE work, and some later papers/open source SAE releases have used this architecture or built on it. Though topK SAEs were a known technique in other contexts, demonstrating they were valuable for interpretability was a real contribution.
* Explored creative ways of measuring SAE performance - the standard methods that we eg used in [Rajamanoharan et al](https://arxiv.org/abs/2404.16014) (interpretability via max activating dataset examples, and sparsity-reconstruction trade-offs) are useful, but only just proxies. The methods used in this paper are also just proxies, of course, but I appreciate that a broader range was explored. In particular, sparse probing is a very reasonable approach that I hadn't seen used before with SAEs - make a dataset for a concept we expect to be learned, and see if it's in there

(Disclosure: I was not at all involved in this paper, but I do know the authors and likely have some positive bias due to that. No one asked me to write this)

(Note also: I don't feel confident in the norms here, so feel free to ignore this comment if these kinds of thoughts are not welcome! I don't see other researchers doing this, but I figure that as ICLR has made a deliberate choice to allow public comments during the rebuttal process, they likely want this kind of public feedback from uninvolved researchers, so long as it's constructive)

---

> ### Public Comment · ~Johan_Edstedt1 · 2024-11-20
>
> This basically breakes double blind.
> Maybe not a suitable comment to make.

---

> ### Public Comment · ~Naomi_Saphra1 · 2024-11-20
> **????**
>
> I like the paper, but direct appeals to the referees from high-profile allies cannot be part of the review process. If public discussion opens the ACs to Oscar-style award campaigns, then ICLR shouldn't allow comments at all until after decisions are finalized.

---

### Comment · Area_Chair_heUp · 2024-11-25
**Author Reviewer Discussion**

Dear Reviewers,

Thank you for your efforts in reviewing this paper. We highly encourage you to participate in interactive discussions with the authors before November 26, fostering a more dynamic exchange of ideas rather than a one-sided rebuttal.

Please feel free to share your thoughts and engage with the authors at your earliest convenience. At this point, I would also encourage you to focus on the author response and on how this addresses your question.

Thank you for your service for ICLR 2025.

Best regards,

AC

---

### Meta-Review · Area_Chair_heUp · 2024-12-21

**Metareview:**

The paper proposes scaled sparse autoencoders with top-k activation to directly control the number of active latent variables and thereby improve interpretability. The proposed approach bares significant benefits in terms of reconstruction ability, non-dead latents and sparsity, fostering improvements in terms of model interpretability. Further, the model is scaled to GPT4 to for practically relevant model analysis. A thorough ablation is provided on GPT-2.
The paper has three very supportive reviews (score 10), highlighting the originality and significance of the work, which has already been adapted in several follow-up works. The paper is very well written, the method description is very detailed and the contributions in terms of the provided training recipe and scaling laws for autoencoders a highly relevant contribution. Technically, the practical scaling of the approach to GPT-4 is also impressive. Overall, the AC agrees that the paper should be highlighted.

**Additional Comments On Reviewer Discussion:**

The paper has received three very supportive reviews. However, reviewer CUAP only gives a score of 3, pointing out limitations in terms of originality and potential practical restrictions through the topK activations. I have read the respective discussion, the authors response and the referred paper and agree with the other reviewers on that matter.
The public comments have not been taken into account for the final decision.

---

### Decision · Program_Chairs · 2025-01-22

Accept (Oral)